# On the non-stationarity of hydrological response in anthropogenically unaffected catchments: An Australian perspective

Hoori Ajami[1,2], Ashish Sharma[1], Lawrence E. Band[3], Jason P. Evans[4], Narendra K. Tuteja[5], G E Amirthanathan[6],
Mohammed A. Bari[7]

[1]School of Civil and Environmental Engineering, University of New South Wales, Sydney, Australia

[2]Department of Environmental Sciences, University of California Riverside, Riverside, USA

[3]Department of Geography and Institute for the Environment, University of North Carolina, Chapel Hill, USA

[4]Climate Change Research Centre, University of New South Wales, Sydney, Australia

[5]Environment and Research Division, Bureau of Meteorology, Canberra, Australian Capital Territory, Australia

[6]Environment and Research Division, Bureau of Meteorology, Melbourne, Victoria, Australia

[7]Environment and Research Division, Bureau of Meteorology, Perth, Western Australia, Australia

*Correspondence to*: Hoori Ajami (hoori.ajami@ucr.edu)

**Abstract.** Increases in greenhouse gas concentrations are expected to impact the terrestrial hydrologic cycle through changes in radiative forcings and plant physiological and structural responses. Here we investigate the nature and frequency of non-stationary hydrological response as evidenced through water balance studies over 166 anthropogenically unaffected catchments in Australia. Non-stationarity of hydrologic response is investigated through analysis of long term trend in annual runoff ratio (1984-2005). Results indicate that a significant trend ($p < 0.01$) in runoff ratio is evident in 20 catchments located in three main ecoregions of the continent. Runoff ratio decreased across the catchments with non-stationary hydrologic response with the exception of one catchment in northern Australia. Annual runoff ratio sensitivity to annual fractional vegetation cover was similar or greater than sensitivity to annual precipitation in most of the catchments with non-stationary hydrologic response indicating vegetation impacts on stream flow. We use precipitation-productivity relationships as the first order control for ecohydrologic catchment classification. Twelve out of 20 catchments present a positive precipitation-productivity relationship possibly enhanced by $CO_2$ fertilization effect. In the remaining catchments, biogeochemical and edaphic factors may be impacting productivity. Results suggest vegetation dynamics should be considered in exploring causes of non-stationary hydrologic response.

**Keywords:** non-stationarity, runoff ratio, catchment classification, vegetation productivity, ecohydrology

## 1 Introduction

Increases in atmospheric $CO_2$ concentration are impacting the terrestrial water cycle through changes in radiative forcings (affecting precipitation and temperature) as well as plant physiological and structural responses (Betts et al., 2007; Wigley and Jones, 1985). As a result, projections of future changes in water resources become complicated due to the tight coupling between the terrestrial biosphere and hydrologic cycle (Band et al., 1996; Baron et al., 2000; Gedney et al., 2006; Ivanov et al., 2008). There is a growing body of evidence showing that increases in $CO_2$ often leads to decreases in leaf stomatal conductance (Field et al., 1995; Medlyn et al., 2001) and lower leaf-scale transpiration rates. However, the impact of reducing stomatal conductance on canopy-scale evapotranspiration (ET) and vegetation productivity (biomass and leaf area index (LAI) increases) is uncertain. In some ecosystems decline in leaf-scale ET rates increases soil available water (Leuzinger and KÖrner, 2010). At the canopy scale, leaf-scale decline of ET might be compensated by increases in plant productivity and changes in ecosystem structure in terms of increases in LAI and changes in species composition (Kergoat et al., 2002). Due to this "compensatory response", the impact of elevated $CO_2$ on catchment scale water balance is uncertain and expected to vary from region to region (Field et al., 1995; Kergoat et al., 2002). Moreover, terrestrial vegetation productivity is often limited by availability of nutrients, mostly nitrogen and phosphorous (Vitousek and Howarth, 1991), and light (Huxman et al., 2004; Schurr 2003) which further increases uncertainty of projecting terrestrial ecosystem response to climate change (Wieder, 2014).

Understanding spatial and temporal variability of catchment scale water yield in relation to precipitation variability and ecosystem productivity is challenging as it requires long term observational records from unimpaired catchments. A plethora of modeling studies have been performed to predict the climate change impacts on vegetation productivity (Kergoat et al., 2002; Leuzinger and KÖrner, 2010) and global runoff (Betts et al., 2007; Piao et al., 2007). However, projections depend on the underlying model assumptions and structure, process representation and scale of application (Medlyn et al., 2011). Similarly, assessing climate elasticity of stream flow has shown that the degree of sensitivity of stream flow to various factors depends on the model structure and calibration approach (Sankarasubramanian et al., 2001).

Here we investigate the nature and frequency of non-stationary hydrologic response as evidenced through water balance studies over 166 anthropogenically unaffected catchments in Australia. Our assessment assumes the non-stationarity to manifest itself through the annual water balance, and more specifically, through the annual runoff ratio (Q/P). Our primary objective is to investigate first whether there is evidence for such non-stationarity in the runoff ratio, and if there is, what could explain its existence.

Non-stationarity of runoff ratio is caused by complex interactions between precipitation, climate variability, plant physiological and structural responses to elevated $CO_2$ (Leuzinger and KÖrner, 2010; Chiew et al., 2014) and landscape characteristics (soil and topography) of a catchment. The question is whether patterns of similarities and differences across space and time exist as a result of these interactions to provide a framework for hydrologic prediction (Sivapalan et al., 2011). Instead of assuming vegetation as a static component of the hydrologic system (Ivanov et al., 2008), catchment scale vegetation dynamics will be an integral component of this classification framework. Therefore, our secondary objective is to formulate a catchment classification framework based on catchment scale ecohydrologic response. This first order grouping of catchments helps to generalize catchment behaviour in terms of changes in runoff ratio and vegetation productivity due to changes in precipitation. Previous catchment classification efforts have mostly considered hydrologic signatures related to precipitation, temperature and stream flow (Sawicz et al., 2011; Wagener et al., 2007). Sawicz et al. (2014) illustrated that changes in climate characteristics of catchments can mostly explain hydrologic change which was characterized by changes in groupings of 314 catchments in the USA. Due to the lack of information, temporal changes in land use were not considered in characterizing hydrologic change in this approach. We argue that in the context of climate change and to improve hydrologic prediction under change (Sivapalan et al., 2011), developing a catchment classification framework that incorporates the role of vegetation dynamics on catchment scale water partitioning is required. This framework can inform future modeling experiments for determining the relative importance of contributing factors to non-stationary catchment response.

An assessment of the non-stationarity of the runoff ratio across 166 anthropogenically unaffected catchments in Australia is presented next using long term ground and satellite-based observational records.

## 2 Data and methods

### 2.1 Data

Daily stream discharge data are obtained from the Australian network of Hydrologic Reference Stations (HRS) that consists of 221 gauging stations (http://www.bom.gov.au/water/hrs/). Out of 221 catchments, 166 catchments have complete daily discharge time series covering the 1979 to 2010 period and these are the catchments used in the study (Fig. 1). The anthropogenically unaffected catchments cover a range of spatial scales with their areas ranging from 6.6 to 232,846 $km^2$. Catchment averaged daily precipitation, actual and potential evapotranspiration, and temperature are obtained from the Australian Water Availability Project (AWAP) gridded time series products at 0.05° resolution (Raupach et al., 2009, 2012). AWAP potential evapotranspiration is calculated based on the Priestley-Taylor equation (Raupach et al., 2009). Monthly fraction of Photosynthetically Active Radiation (fPAR) absorbed by vegetation is obtained from Donohue et al. (2008) at

0.08° resolution for the 1984-2010 period. Total fPAR ($F_{tot}$) values are approximately related to fractional vegetation cover and ranges between 0.0 (no vegetation cover) to 0.95 (maximum vegetation cover). The monthly $F_{tot}$ dataset version 5 are derived from the Advanced Very High Resolution Radiometer (AVHRR) sensor, and $F_{tot}$ values were used as a measure of vegetation productivity in this study assuming energy use efficiency is constant (https://data.csiro.au). This dataset has been previously used to assess trends in vegetation cover across Australia (Donohue et al., 2009).

## 2.2 Methods

We used ground and satellite-based observations to detect and investigate causes of non-stationarity of runoff ratio across HRS catchments. Our methodology consists of: 1) detecting trends in annual runoff ratio, fractional vegetation cover, annual precipitation and precipitation seasonality indices at a catchment scale, 2) assessing long term (27 years) water balance patterns across all catchments with non-stationary hydrologic response using hydrologic indices such as the Horton index (Troch et al., 2009), 3) exploring annual runoff ratio's sensitivities to water balance components and fractional vegetation cover at an individual catchment scale, and 4) formulating an ecohydrologic catchment classification framework.

### 2.2.1 Non-parametric trend analysis to detect non-stationarity

The modified Mann Kendall non-parametric test (Hamed and Rao, 198l; Kendall, 1970; Mann, 1945) that accounts for serial autocorrelation in the time series is performed to detect significant trends in annual runoff ratio at 0.01 significance level across the 166 HRS catchments. The first and last 5 years of data are removed from the record to reduce the impact of edge effect for trend analysis (1984-2005). Similar trend analysis is performed for annual precipitation and average fractional vegetation cover of each catchment.

Changes in precipitation seasonality across catchments with non-stationary hydrologic response are explored by assessing the trends in two measures of precipitation seasonality: the seasonality index (SI) (Walsh and Lawler, 1981) and days of a year at which the 10th, 25th, 50th, 75th and 90th percentiles of annual precipitation are reached (Pryor and Schoof, 2008). The SI is calculated based on monthly precipitation values (Walsh and Lawler, 1981):

$$SI = \frac{1}{P}\sum_{n=1}^{12}\left|X_n - \frac{P}{12}\right| \quad (1)$$

Where P is annual precipitation and $X_n$ is monthly total precipitation in month n.

### 2.2.2 Hydrologic similarity across catchments with non-stationary hydrologic response

Similarity of ecohydrologic response across all catchments with non-stationary response is explored by examining the overall relationships of long term mean (1984-2010) annual fractional vegetation cover with annual runoff ratio, precipitation and the Horton index (Troch et al., 2009). Horton index, ratio of evapotranspiration to catchment wetting, presents efficiency of catchments in using plant available water and is reflective of water and energy availability in the catchment. Horton index ranges between 0 and 1, and incorporates the role of soil and topography in the catchment wetting (Brooks et al., 2011; Troch et al., 2009; Voepel et al., 2011). To estimate catchment averaged ET and wetting, the water balance equation (dS/dt = P – Q - ET) is used assuming that changes in annual storage (dS/dt) is zero.

$$ET = P - Q \tag{2}$$

$$W = P - S, \tag{3}$$

Where P is annual precipitation, Q is the total stream discharge, ET is annual actual evapotranspiration, W is catchment wetting, and S is the quick flow component of stream discharge. A one parameter recursive filter of Lyne and Hollick (1979) for baseflow separation is used to estimate quick flow and baseflow components of daily discharge.

$$b_k = a\, b_{k-1} + \frac{1-a}{2}(Q_k + Q_{k-1}) \tag{4}$$

$$S = Q - b_k \tag{5}$$

Where $b_k$ is baseflow and $a$ is the filter parameter and typically is set to 0.925 (Xu et al., 2012). In arid and semiarid catchments as quick flow constitutes most of the total stream flow, HI is approaching 1. In humid catchments, quick flow runoff is smaller than the total stream flow and HI is less than 1. In catchments with limited storage, HI is undefined (0/0) (Troch et al., 2009). Next, inter-annual variability of catchment scale ecohydrologic response is explored.

**2.2.3 Normalized sensitivities of annual runoff ratio to changes in water balance and vegetation**

Sensitivities of annual runoff ratio to inter-annual variability of precipitation, ET, and fractional vegetation cover in 1984-2010 period are computed to identify factors that exert the largest sensitivity on annual runoff ratio. Normalized sensitivity of annual runoff ratio to precipitation is computed by estimating the slope of a linear regression between runoff ratio and precipitation, and multiplying it by the ratio of mean precipitation to runoff ratio (Fatichi and Ivanov, 2014; Hsu et al., 2012). Similarly, normalized sensitivity of runoff ratio to water balance ET and annual fractional vegetation cover are also computed. Normalized sensitivity of annual runoff ratio is equivalent to the stream flow elasticity approach of Zheng et al. (2009) that defined stream flow elasticity as the linear regression coefficient between the proportional changes in stream flow and a climatic variable (precipitation or potential evapotranspiration). Results of these analyses are used as the basis for formulating an ecohydrologic catchment classification.

**3 Results**

### 3.1 Non-stationary hydrologic response

Results of the modified Mann-Kendall trend test across 166 catchments indicate that 20 catchments (areas range between 18.7 to 2677 km$^2$, Table 1) have significant decreasing trends in annual runoff ratio ($p < 0.01$) except for the East Baines River in northern Australia (Fig. 1). An increasing trend for runoff ratio in the East Baines River (0.009/yr) is consistent with annual precipitation increases (13.2 mm/yr). Moreover, this catchment has the smallest fractional vegetation cover (0.26) amongst the catchments with non-stationary hydrologic response. The North Esk catchment in Tasmania is the only catchment amongst the catchments with non-stationary response in which runoff ratio declined despite increases in annual precipitation (6.4 mm/yr) (Table 1). In the Tasmanian catchment, the increasing trend in fractional vegetation cover (0.009/yr) is significant and results in ET increase and subsequently lower runoff ratio during 1984-2005 period. In the rest of the catchments with non-stationary hydrologic response, total annual precipitation decreased between -1.9 mm/yr to -24.7 mm/yr in 1984-2005 period which is consistent with the decreasing trend in annual runoff ratio (-0.0008/yr to -0.016/yr). However, most catchments present an increasing trend in annual fractional vegetation cover with the exception of three catchments in the Eastern Australia Temperate forests (410705, 410761, and 412066). Tree is the dominant vegetation cover in all the catchments with non-stationary hydrologic response except for the Avoca River at Coonooer (408200) and Mollison creek (405238) catchments in Victoria in which grasslands are dominant (Table S1). The question is what causes the increasing trends in annual fractional vegetation cover despite decreasing trends in annual precipitation of these catchments?

Based on the mean seasonality index using data from 1984 to 2010 period, only two catchments exhibit a seasonal climate (0.6 < SI < 0.8) (Table S2). However, all catchments have some degree of rainfall seasonality (SI > 0.39) (Walsh and Lawler, 1981). Using the modified Mann-Kendall trend tests, no significant trends in the 1984-2005 SI values are observed in the catchments with non-stationary hydrologic response ($\alpha=0.01$). Few significant trends in precipitation seasonality indices using the percentiles are observed in the catchments with non-stationary hydrologic response including in catchments 410061, 410731, 405238 and 212260. In these catchments significant trend in the timing of the 25th percentile are detected except for catchment 212260 where the trend was significant for the 50th percentile. The seasonal shifts in precipitation can impact vegetation dynamics particularly when they occur between the growing and non-growing seasons such as in catchment 410731. This result suggests that other factors besides precipitation are contributing to observed non-stationarity. Next, we explore hydrologic similarity across catchments with non-stationary hydrologic response.

### 3.2 Long term patterns of hydrologic similarity across catchments with non-stationary hydrologic response

Long-term annual average dryness index (PET/P) (1984-2010) of the 166 study catchments illustrates presence of energy- and water-limited catchments in the region (Fig. S1). The North Esk catchment in Tasmania is the only energy-limited catchment amongst the catchments with non-stationary response. Across the catchments with non-stationary hydrologic response increases in mean annual precipitation (1984-2010) increases mean annual fractional vegetation cover particularly in catchments with mean annual precipitation of less than 800mm (Fig. 2). After that, increases in $F_{tot}$ reach an asymptote with mean annual precipitation greater than 800. Similarly, runoff ratio and its variability increases due to precipitation increase particularly across catchments with mean annual precipitation of greater than 800 mm and mean annual fractional vegetation cover of greater than 0.7. The exception is the East Baines River in northern Australia which has the smallest fractional vegetation cover but has large variability in runoff ratio. In drier catchments, Horton index is close to 1 and exhibits smaller variability compared to the wetter catchments.

To explore differences between catchments with non-stationary or stationary behaviour, the cumulative absolute differences between consecutive annual values of precipitation, fractional vegetation cover and runoff ratio for each catchment are calculated and normalized by the total absolute difference. In Figure 3, the differences between catchments with non-stationary and stationary hydrologic response are illustrated by presenting the mean and standard deviations of normalized cumulative differences for each group. As can be seen in Figure 3, normalized cumulative differences in annual precipitation and fractional vegetation cover between the catchments with non-stationary and stationary hydrologic response are very similar. However, large differences in the normalized cumulative differences of annual runoff ratio exist between these catchments. The catchment area ranges from 18.7 to 5158 km$^2$ in catchments with non-stationary hydrologic response (Table 1). While increases in runoff ratio, P-Q and mean fractional vegetation cover with increases in mean catchment slope are observed in catchments with non-stationary hydrologic response, no distinct differences between catchments with stationary and non-stationary hydrologic response are observed.

Although, consistent patterns are observed in catchments' ecohydrologic response due to differences in mean annual precipitation in catchments with non-stationary hydrologic response, characterizing catchment scale terrestrial ecosystem response to inter-annual precipitation variability is important for hydrologic predictions.

### 3.3 Normalized sensitivities of annual runoff ratio at a catchment scale

Normalized sensitivity of annual runoff ratio to annual fractional vegetation cover, ET and precipitation, indicate greater sensitivity of runoff ratio to fractional vegetation cover than precipitation in most of the catchments with non-stationary hydrologic response (Fig. 4a). While runoff ratio's sensitivities to precipitation are positive across all catchments with non-

stationary hydrologic response, these sensitivities become negative in some catchments with increases in fractional vegetation cover. These results indicate the importance of incorporating vegetation dynamics in examining non-stationary hydrologic response.

Normalized sensitivity of annual fractional vegetation cover to precipitation (Fatichi and Ivanov, 2014, Hsu et al., 2012) is plotted against mean aridity index (PET/P) (Fig. 4b). As can be seen in Figure 4(b), fractional vegetation cover presents both positive and negative sensitivities to precipitation inter-annual variability. Across catchments with positive precipitation-fractional vegetation cover relationships, fractional vegetation cover sensitivities approach zero in catchments with aridity index of 1.5. Fractional vegetation cover sensitivity is highest in the semi-arid catchments with lower mean annual

precipitation compared to the rest of the catchments with non-stationary hydrologic response. In catchments with mean annual precipitation of 800 mm or higher (aridity index < 1.5), slopes of fractional vegetation cover to mean annual precipitation are zero or negative. A negative slope indicates increases in fractional vegetation cover despite precipitation decrease.

As vegetation productivity is controlled by plant available water (Brooks et al., 2011), fractional vegetation cover sensitivity to the Horton index is explored. Both positive and negative sensitivities between the fractional vegetation cover and Horton index are observed in catchments with non-stationary hydrologic response (Fig. 4c). Positive sensitivities indicate increases in fractional vegetation cover as the Horton Index increases. As higher Horton index is indicative of a drier condition, removal of limiting factors like a nutrient limitation is the likely cause of fractional vegetation cover increase in these

catchments (Brooks et al., 2011). In a few of these catchments, light limitation may decrease fractional vegetation cover in wet years (positive correlations of sunshine hours with fractional vegetation cover, Table S3). In catchments with negative Horton index-fractional vegetation cover sensitivities in which drier conditions decrease productivity, water availability is the primary factor in controlling vegetation growth. Annual runoff ratio's sensitivity to fractional vegetation cover was similar to the Horton index but with the opposite sign (Fig. 4d). Across water limited catchments (positive runoff ratio-

fractional vegetation cover relationship), runoff ratio's sensitivities are smallest in catchments with the highest vegetation cover. As periods of higher productivity coincide with higher precipitation (positive precipitation-fractional vegetation cover relationship) in these catchments, runoff ratio increases in years with higher precipitation. It should be noted that the percentage of tree cover in these drier catchments are more than 60% with a few exceptions (Table S1). Negative runoff ratio-fractional vegetation cover sensitivities become more negative in catchments with higher fractional vegetation cover.

Overall, mean annual runoff ratio and its variability (standard deviation) are smaller in drier catchments with smaller mean fractional vegetation cover (Fig. 2). We used baseflow as a measure of catchment storage response to inter-annual

precipitation variability. Baseflow's sensitivities to mean annual aridity index are highest in drier catchments with non-stationary hydrologic response (Fig. 5a). Normalized fractional vegetation cover sensitivities to the baseflow decrease in catchments with higher annual baseflow index and even become negative at higher baseflow indices (Fig. 5b). This result suggests that in catchments where groundwater constitutes significant component of stream flow, fractional vegetation cover exhibits smaller variability to changes in baseflow as vegetation roots have access to deeper water storage for transpiration and have less sensitivity to changes in baseflow.

Consistent patterns of fractional vegetation cover sensitivities to precipitation, baseflow, and Horton Index across catchments with non-stationary hydrologic response present two distinct catchment response behaviours. We hypothesize plausible mechanisms to describe the likely causes of fractional vegetation cover sensitivity to inter-annual precipitation variability in order to distinguish between alternate catchment ecohydrologic responses.

### 3.4 Formulating catchment scale ecohydrologic response

At the global scale precipitation is the main driver of vegetation productivity particularly in arid and semi-arid environments (Huxman et al., 2004). However, mean annual vegetation productivity becomes less sensitive to mean annual precipitation in humid environments (Schuur, 2003) as biogeochemical factors (nutrients, light, soil oxygen availability) or biotic factors (Yang et al., 2008) limit productivity (Fig. 6a). This is consistent with observed precipitation-fractional vegetation cover pattern across all the catchments with non-stationary hydrologic response (Fig 2a.). At a catchment scale, catchments can be classified into two main groups based on the annual precipitation and vegetation productivity relationship. We hypothesize four plausible mechanisms to explain catchment scale ecohydrologic response to inter-annual climate variability in water and energy-limited environments (Fig. 6b).

In group (A) catchments, a positive relationship between vegetation productivity and precipitation increases exists and can be either caused by 1) direct $CO_2$ fertilization effect in which increases in $CO_2$ enhances photosynthesis and increases LAI, and ET will increase due to precipitation and LAI increase (Fig. 6b - class A1), or by 2) indirect $CO_2$ fertilization effect in which increased $CO_2$ gradient between the atmosphere and leaf enhances photosynthesis but LAI does not increase. Therefore, reduction in the stomatal conductance reduces ET (Fig. 6b - class A2) (Ainsworth and Long, 2005). In group (A) catchments, changes in runoff ratio depend on the hydroclimatic condition. In years where precipitation increase is higher than ET, increase in productivity is followed by increases in runoff ratio while in drier than average years increases in ET reduce the runoff ratio. It is expected that under future warming, $CO_2$ increases will continue to increase productivity unless

decreases in plant water availability limit plant growth, or changes in stomatal conductance, plant respiration rates (Wu et al., 2011), and nutrient availability impact productivity.

In group (B) catchments, vegetation productivity decreases in response to annual precipitation increases. This negative feedback is most likely due to biogeochemical constraints such as light, nutrients, temperature and soil characteristics (Bai et al., 2005) despite changes in the stomatal conductance due to $CO_2$ increases (Paruelo et al., 1999). In these catchments, productivity is likely constrained by the 1) nutrients (class B1) or 2) light availability (class B2). In catchments where increases in precipitation are followed by ET increases, nutrient limitation (Norby et al., 2010, Schuur, 2003) is the likely cause of decline in productivity (Fig 6b - class B1). In B2 catchments light and other factors (anoxic conditions, temperature) limit productivity and decline ET despite increases in precipitation. As these catchments are in the wetter regions, nutrient limitation might be caused by increased nutrients leaching in wet soils (Schlesinger, 1997) or increases in nutrient-use efficiency due to water availability which subsequently leads to nutrient limitation (Paruelo et al., 1999). Similar to group (A), changes in runoff ratio depend on the catchment's hydroclimatic condition. In these catchments future changes in vegetation productivity is likely dependent on the rate of nutrient mineralization (Brooks et al., 2011), nitrogen deposition and changes in disturbance regimes such as fire and drought. A flowchart illustrates how catchment classification is performed by computing Spearman rank correlations between two variables at each step (Fig. 6c).

The prevalence of the four classes identified above, is presented using time series of annual precipitation, water balance derived ET and runoff ratio as well as catchment averaged fractional vegetation cover for the 1984-2010 period. Three constitutive relationships are established for every catchment at an annual scale between: 1) precipitation 2) runoff ratio and 3) ET versus catchment averaged fractional vegetation cover. Catchment scale transpiration data are not available for this classification. According to these relationships and Spearman rank correlations, catchments with non-stationary hydrologic response are grouped in three classes (A1, B1 and B2, Fig. 1). None of the catchments with non-stationary hydrologic response presented a relationship proposed for class A2 catchments. Figure 7 show Spearman rank correlation values for example catchments in each class.

As presented in Figure 1 and Table 2, 12 catchments are classified as class A1. The Spearman rank correlations between annual precipitation and fractional vegetation cover in class A1 catchments are positive and typically larger than class B1 and B2 catchments, and in 8 out of 12 catchments the correlation is significant ($p < 0.05$). Only one catchment in class B1 (total 3) has a significant negative correlation between precipitation and fractional vegetation cover.

While data on catchment scale nutrient availability are not available, general ET-fractional vegetation cover relationships in group B catchments can be further explained by annual precipitation-temperature relationships. In wetter years despite lower vegetation cover, ET will likely increase due to higher water availability in warmer years in B1 catchments (positive precipitation-temperature correlations) (Table S3). In B2 class with negative precipitation-temperature relationships, cooler temperatures and light limitation decline ET.

Groupings of all A1, and B1 and B2 catchments illustrate significant correlations for all three constitutive relationships of Figure 7 ($p < 0.05$, Table 2) in group A1 and group B catchments except between fractional vegetation cover and ET. Therefore, precipitation-fractional vegetation cover relationships present first order groupings of the catchments. Further distinction within a group is speculative as it depends on catchment derived annual ET.

## 4 Discussions

According to our analysis, catchments with non-stationary hydrologic response present three distinct behaviours as a result of inter-annual variability in catchment water balance and vegetation fractional cover. In the following, we discuss whether the proposed catchment classification is consistent once other measures or data are used.

### 4.1 Did catchments with non-stationary hydrologic response experience similar changes in vegetation and water balance variables?

To explore whether HRS catchments are undergone similar changes during the period of analysis, regime curves based on daily runoff, precipitation and monthly fractional vegetation cover data for each catchment are developed using data from pre-drought (1984-1996) and drought period (1997-2009) (Coopersmith et al., 2014). Regime curves are obtained by averaging daily values of precipitation or runoff for a given day over the length of the data. As daily fractional vegetation cover data are not available, monthly values are used to develop the regime curves. To summarize the differences between the regime curves for the pre-drought and drought periods, Nash Sutcliffe Efficiency (NSE) criterion is calculated. As can be seen in Figure 8, differences in daily precipitation and runoff and monthly fractional vegetation cover regime curves are much higher (indicated by negative NSE) in catchments with non-stationary hydrologic response than the catchments that do not exhibit non-stationary behaviour. While the results of trend analysis are impacted by defining the significance level, the above analysis indicates that catchments with non-stationary behaviour have undergone larger changes. To further assess the impact of significance level on the results of the trend analysis, the approach of Douglas et al. (2000) for computing the field significance of regional trend tests are implemented. In this approach, time series of runoff ratio for every catchment are resampled 10,000 times using the bootstrap approach. In the next step, the Kendall's S is calculated for each bootstrap sample and the regional test statistics is calculated by averaging Kendall's S for each iteration and computing non-

exceedance probability using the Weibull plotting position formula. Finally, the CDF of regional test statistics is compared with the historical Kendall's S calculated for each station using 0.01 significance level. Indeed, the field significance level obtained from the bootstrap samples is 0.0239 which is more relaxed than the p-value = 0.01 originally used. Using the new field significance level, 34 catchments are classified as non-stationary.

## 4.2 Is the ecohydrologic catchment classification consistent across other measures?

Positive precipitation-fractional vegetation cover relationships in class A1 catchments is consistent with positive normalized fractional vegetation cover's sensitivities of individual catchments to annual precipitation (Fig. 4b and S2) and indicate that water availability primarily controls fractional vegetation cover increase in A1 catchments. A positive Spearman rank

correlation between the coefficient of variation (CV) of annual fractional vegetation cover and CV of annual precipitation (r = 0.34, $p$ = 0.3) across all A1 catchments further confirms this conclusion (Yang et al., 2008).

In group B catchments, negative normalized sensitivities of fractional vegetation cover to precipitation exist (Fig. 4b). This pattern is followed by a negative correlation between the CVs of these two factors across all group B catchments (r = -0.71, $p$

= 0.06) which highlights the role of biogeochemical factors in controlling productivity. Small or even negative sensitivities of vegetation cover to precipitation in group B might be due to the presence of perennial vegetation (shrubs and trees) as ecosystems with more perennial cover are less responsive to inter-annual precipitation variability (Jin and Goulden, 2014).

Our classification framework suggests that class A catchments are more sensitive to increases in $CO_2$ concentrations than the

class B catchments that are in the humid zone (P/PET > 0.65). This result is consistent with Ukkola et al. (2016) as they showed greater sensitivities of annual ET and NDVI to increases in $CO_2$ concentrations in sub-humid and semi-arid catchments of Australia.

## 4.3 Are the inferred classification patterns artefacts of remote sensing data and catchment scale ET?

To assess whether observed precipitation-productivity relationships are the artefacts of remote sensing data, two independent remote sensing vegetation products are used: Vegetation Optical Depth (VOD), and Enhanced vegetation index (EVI) (Huete et al., 2002; 2006). Global long-term (1988-2010) annual Vegetation Optical Depth (VOD) dataset from passive microwave satellites with 0.25° resolution (Liu et al., 2011) is related to water content of leaf and woody components of aboveground biomass (Liu et al., 2015), and is able to detect structural differences in areas with near-closed canopy. Spearman rank

correlations between VOD and annual precipitation across group B catchments were negative and consistent with the results of AVHRR fractional vegetation cover data (Text S2, Table S4). Moreover, the Australia coverage of despiked EVI dataset

(2001-2010) from the Moderate Resolution Imaging Spectroradiometer (MODIS) presented high correlations with $F_{tot}$ data (2001-2010). Previous investigations have shown that EVI is more sensitive to net primary productivity compared to the normalized vegetation index (Huete et al., 2002; 2006). Analyses from these two independent datasets reduce uncertainty of identifying negative precipitation-productivity correlations at the catchment scale. However, further research is required to
determine exact causes of the observed behaviour.

Here, we assumed that changes in catchment storage at the annual scale are zero to compute annual water balance ET. However, this assumption is likely not correct in all years. Using AWAP actual annual ET similar relationships between fractional vegetation cover and annual ET are obtained except for three catchments in group B2 (Table 2). AWAP ET is
based on daily transpiration and soil evaporation values obtained from the WaterDyn model that simulates terrestrial water balance across Australia at 5 km resolution (Raupach et al., 2009). In addition to inter-annual water storage carry over, inter-annual non-structural carbon storage across years (a wet year can result in greater biomass/leaf area in the following year) can impact precipitation-vegetation relationships.

**4.4 Does precipitation-fractional vegetation cover relationship depend on the period of analysis?**
The period of analysis is limited to 1984-2010 in this study due to availability of AVHRR fractional vegetation cover data for Australia. To assess sensitivity of precipitation-fractional vegetation cover relationships to data length and catchment condition, these relationships are developed for two time periods: 1984-1996 and 1997-2009. It should be noted that 1997-2009 corresponds to the millennium drought in Australia (Chiew et al., 2014). Results indicate similar precipitation-
fractional vegetation cover relationships to 1984-2010 in class A1 as well as in class B with a few exceptions (Fig. S2). Despite these exceptions, the drier conditions of 1997-2009 resulted in higher mean fractional vegetation covers in group B compared to 1984-1996 period consistent with the classification framework. Results suggest that the record length is important in catchments where productivity is limited by resources besides water availability.

**5 Summary and conclusions**
We used precipitation-fractional vegetation cover relationships for first order groupings of catchment scale ecohydrologic response in 20 catchments with non-stationary hydrologic response located in different hydroclimatic regions of Australia. Our results illustrate that fractional vegetation cover is more sensitive to increases in precipitation (stronger Spearman rank correlations) in class A1 catchments (12 catchments). This inference is consistent with the result of meta-analysis of
productivity response to precipitation across the globe (Wu et al., 2011). The drawback of using precipitation as the main driver of vegetation productivity is that the impact of confounding variables that co-vary with precipitation is ignored (Wu et

al., 2011). Fractional vegetation cover sensitivity to precipitation and Horton index provided consistent results with our catchment classification framework except for two catchments. These catchments (408202, 410061) have smaller rank correlation between precipitation and fractional vegetation cover compared to the rest of class A1 catchments. Eight out of 20 catchments with non-stationary hydrologic response present negative precipitation-fractional vegetation cover relationships impacted by nutrient or light availability.

While, determining the exact causes of non-stationarity requires detailed modeling experiments, non-stationarity of runoff ratios could be attributed to changes in precipitation amount, intensity and seasonality, increases in air temperature and $CO_2$ concentrations (Chiew et al., 2014). The proposed framework provides a general guideline for projecting the likely changes in catchment water balance in response to climate change and designing simulation experiments. However, uncertainty still remains about the terrestrial ecosystem response as factors such as nutrients and light availability, vegetation developmental stage, space constraint and prevalence of pests may impact productivity (Körner, 2006). In addition, it is expected that frequency and duration of extreme events such as fire, drought and floods, will increase which can further alter ecosystem response and plant water availability (Medlyn et al., 2011).

**Acknowledgements**

This research was funded by the Australian Research Council linkage grant (LP130100072), the Australian Bureau of Meteorology and WaterNSW. We acknowledge the Australian Bureau of Meteorology for providing the Hydrologic Reference Station data supported by the Australian Government through the Water Information Program. We would like to acknowledge Dr. Yi Liu for providing the VOD data.

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

**Table 1:** Mean annual precipitation (P), discharge (Q), runoff ratio (Q/P) and fractional vegetation cover ($F_{tot}$) of catchments with non-stationary hydrologic response during 1984-2005 period. Slopes of the trend lines obtained from a linear regression model fitted to each time series.

| Station | Area (km²) | Mean P (mm) | Mean Q (mm) | Mean Q/P (-) | Mean $F_{tot}$ | Slope of the trend (1984-2005) Q/p | P | $F_{tot}$ |
|---|---|---|---|---|---|---|---|---|
| 1. 212260 | 713 | 886.8 | 189.4 | 0.19 | 0.75 | -0.013[*] | -13.3[*] | 0.0052[*] |
| 2. 215002 | 1382.2 | 803.2 | 155.6 | 0.17 | 0.72 | -0.012[*] | -15.5[*] | 0.0041 |
| 3. 215004 | 165.6 | 935.8 | 292.5 | 0.29 | 0.69 | -0.011[*] | -15.6 | 0.0049 |
| 4. 216004 | 95.7 | 1125.8 | 204.3 | 0.16 | 0.71 | -0.010[*] | -24.7[*] | 0.0048 |
| 5. 218001 | 90.6 | 815.1 | 266.9 | 0.3 | 0.72 | -0.016[*] | -10 | 0.0008 |
| 6. 406214 | 237 | 580.3 | 45 | 0.07 | 0.54 | -0.005[*] | -6.9 | 0.0018 |
| 7. 408200 | 2677.3 | 508.5 | 6.5 | 0.01 | 0.48 | -0.0008[*] | -6.4 | 0.0014 |
| 8. 408202 | 82.6 | 605 | 48.1 | 0.07 | 0.7 | -0.006[*] | -4.9 | 0.005 |
| 9. 410061 | 146.1 | 1004 | 245.1 | 0.24 | 0.77 | -0.008[*] | -10.5 | 0.0029[*] |
| 10.410705 | 508.6 | 744.8 | 65.1 | 0.08 | 0.64 | -0.006[*] | -11.5 | -0.002 |
| 11.410731 | 671.6 | 897.8 | 84.9 | 0.09 | 0.64 | -0.006[*] | -11.9 | 0.0003 |
| 12.410734 | 563.7 | 816.4 | 93.6 | 0.1 | 0.67 | -0.008[*] | -13.2[*] | 0.0018 |
| 13.410761 | 5158.3 | 742.2 | 56.7 | 0.07 | 0.59 | -0.005[*] | -7.7 | -0.0002 |
| 14.412028 | 2630.7 | 778.6 | 97.2 | 0.11 | 0.69 | -0.006[*] | -12 | 0.0007 |
| 15.412066 | 1629.7 | 785.9 | 100.1 | 0.12 | 0.68 | -0.008[*] | -12 | -0.0002 |
| 16.415207 | 304.5 | 645.7 | 52.2 | 0.07 | 0.66 | -0.006[*] | -6.4 | 0.0013 |
| 17.613146 | 18.7 | 1019.4 | 209.6 | 0.2 | 0.69 | -0.006[*] | -1.9 | 0.0086[*] |
| 18.G8110004 | 2443.1 | 811.7 | 128.1 | 0.15 | 0.26 | 0.009[*] | 13.2 | 0.0015 |
| 19.318076 | 379.8 | 1156.9 | 383.3 | 0.33 | 0.7 | -0.005[*] | 6.4 | 0.0090[*] |
| 20.405238 | 164.1 | 734.5 | 112 | 0.14 | 0.64 | -0.008[*] | -7.7 | 0.0035 |

[*]The trend is significant at 0.01 significance level using the modified Mann-Kendall trend test.

**Table 2:** Catchment properties and Spearman rank correlations (r) for catchments with non-stationary hydrologic response in Australia. Data spans 1984-2010 period. Class categories refer to the catchment classification framework of Figure 6.

| Station | Mean P (mm) | Mean Q (mm) | $r_1$ | $r_2$ | $r_{2\text{-AWAP}}$ | $r_3$ | Class |
|---|---|---|---|---|---|---|---|
| 406214 | 579.7 | 42.8 | 0.41* | 0.47* | 0.43* | 0.21 | A1 |
| 408200 | 501.2 | 5.9 | 0.65* | 0.65* | 0.47* | 0.47* | A1 |
| 408202 | 594.5 | 43.4 | 0.11 | 0.17 | 0.19 | 0.01 | A1 |
| 410705 | 734.9 | 58.1 | 0.61* | 0.63* | 0.70* | 0.55* | A1 |
| 410731 | 882.7 | 75.2 | 0.17 | 0.24 | 0.29 | 0.06 | A1 |
| 410761 | 729.6 | 51.0 | 0.45* | 0.52* | 0.47* | 0.36 | A1 |
| 412028 | 762.1 | 84.8 | 0.47* | 0.53* | 0.40* | 0.29 | A1 |
| 412066 | 774.4 | 88.9 | 0.55* | 0.56* | 0.41* | 0.34 | A1 |
| 415207 | 632.5 | 46.8 | 0.25 | 0.17 | 0.22 | 0.29 | A1 |
| G8110004 | 841.1 | 144.4 | 0.45* | 0.23 | 0.41* | 0.56* | A1 |
| 410061 | 979.1 | 221.2 | 0.06 | 0.18 | 0.38 | -0.09 | A1 |
| 405238 | 723.0 | 100.7 | 0.4* | 0.4* | 0.46* | 0.29 | A1 |
| 215004 | 915.3 | 273.3 | -0.42* | -0.12 | -0.42* | -0.38* | B1 |
| 216004 | 1099.5 | 179.0 | -0.27 | -0.01 | -0.24 | -0.39* | B1 |
| 218001 | 807.5 | 246.6 | -0.23 | -0.07 | -0.15 | -0.16 | B1 |
| 212260 | 876.8 | 175.3 | -0.30 | 0.20 | -0.27 | -0.53* | B2 |
| 215002 | 789.3 | 136.7 | -0.01 | 0.29 | 0.06 | -0.13 | B2 |
| 410734 | 808.8 | 83.9 | -0.03 | 0.17 | 0.2 | -0.09 | B2 |
| 613146 | 990.1 | 188.3 | -0.23 | 0.22 | -0.39* | -0.65* | B2 |
| 318076 | 1151.3 | 375.6 | -0.01 | 0.29 | -0.03 | -0.46* | B2 |
| Class A1 | | | 0.34* | 0.32* | 0.41* | 0.33* | |
| Class B1 | | | -0.23* | 0.008 | -0.13 | -0.28* | |
| Class B2 | | | -0.09 | 0.11 | 0.14 | -0.16 | |
| Class B | | | -0.14* | 0.07 | 0.04 | -0.21* | |

*Correlation is significant ($p < 0.05$); $r_1$: correlation between mean annual fractional vegetation cover and annual precipitation; $r_2$: correlation between annual evapotranspiration (water balance approach) and mean annual fractional

vegetation cover; $r_{2\text{-AWAP}}$: correlation between mean annual fractional vegetation cover and AWAP annual evapotranspiration; $r_3$: correlation between annual runoff ratio (Q/P) and mean annual fractional vegetation cover.

**Figure captions:**

**Figure 1:** Distribution of hydrologic reference stations across Australia. Coloured circles represent catchments with significant trend in annual runoff ratio. Colours represent catchment grouping based on the classification framework of Figure 6. The catchments with non-stationary hydrologic response span over three significant ecoregions of the continent. Ecoregion boundaries are from the World Wildlife Fund (http://maps.tnc.org/gis_data.html).

**Figure 2:** Mean and standard deviation of catchment averaged a) annual fractional vegetation cover, and b) annual runoff ratio, against mean annual precipitation (P) in catchments with non-stationary hydrologic response. c) mean and standard deviation of annual runoff ratio, and d) Horton Index versus catchment averaged annual fractional vegetation cover.

**Figure 3:** Mean normalized cumulative absolute differences in annual precipitation, fractional vegetation cover and runoff ratio between catchments with non-stationary (20 catchments) and stationary (146 catchments) hydrologic response. The shaded areas represent standard deviation.

**Figure 4:** a) Normalized sensitivities of runoff ratio (RR) to precipitation (P), water balance ET and fractional vegetation cover (Ftot), normalized sensitivities of annual: b) fractional vegetation cover to precipitation against catchments' mean aridity index, c) fractional vegetation cover to Horton index (HI) against catchments' mean Horton Index, and d) runoff ratio to fractional vegetation cover against catchments' mean fractional vegetation cover in catchments with non-stationary hydrologic response. Data labels refer to the station identification number in Table 1.

**Figure 5:** a) Normalized baseflow (B) sensitivities to annual precipitation (P) in each catchment with non-stationary hydrologic response against its mean aridity index (1984-2010), b) normalized annual fractional vegetation cover ($F_{tot}$) sensitivities to annual baseflow of each catchment with non-stationary hydrologic response against its mean baseflow index (BFI) (1984-2010). Data labels refer to the station identification number in Table 1. In general, annual baseflow sensitivities to mean annual precipitation decreases in wetter catchments (smaller aridity index). Positive sensitivities of fractional vegetation cover to baseflow decreases in catchments with higher baseflow index. Negative fractional vegetation cover sensitivities to baseflow become more negative in catchments with higher baseflow index indicating larger contribution of groundwater to stream flow.

**Figure 6:** a) Global pattern of annual productivity ($F_{tot}$) and mean annual precipitation relationship. While precipitation is the primary factor for vegetation growth in water limited sites, productivity reaches an asymptote in humid areas or decreases (e.g. tropical forests) with increases in precipitation due to biogeochemical or edaphic constraints. The grey region corresponds to catchments in which productivity is insensitive to inter-annual precipitation variability. b) A conceptual framework for characterizing changes in runoff ratio to changes in annual precipitation and vegetation productivity ($F_{tot}$) in relation to catchment's hydroclimatic condition. In group (A) catchments a positive relationship between annual precipitation and productivity exists and annual ET changes in relation to productivity depend on the dominance of structural control (increases in LAI, class A1) versus physiological control (decreases in stomatal conductance, class A2) in controlling productivity. In group (B) catchments, an inverse relationship between precipitation and productivity exists and productivity is likely constrained by biogeochemical factors. In B1 catchments, negative ET and productivity relation indicates productivity is likely controlled by nutrients availability as drier conditions induce nutrient mineralization. In B2 catchments, light availability and lower temperature reduce ET. In group (A) catchments, runoff ratio would increase as productivity increases while in group (B), runoff ratio will likely decrease with increasing productivity (decreases in precipitation). Depending on the dominance of limiting resource, precipitation-productivity may shift between the two regimes. c) The flowchart illustrates the classification procedure. The classification starts by assessing the correlations between annual precipitation and $F_{tot}$ and then annual ET and $F_{tot}$ in a catchment.

**Figure 7:** Relationships between catchment averaged annual fractional vegetation cover and annual precipitation (left), water balance derived annual ET (middle) and runoff ratio (right) against mean annual fractional vegetation cover (1984-2010) across three catchments representative of each class in Figure 6. The Spearman rank correlation (r) and p-values are shown when correlation is significant.

**Figure 8:** Box plots of NSE values calculated between the regime curves of pre-drought and drought periods in catchments with non-stationary (20 catchments) and stationary (146 catchments) hydrologic response respectively. Changes in daily precipitation and runoff and monthly fractional vegetation cover were larger in catchments with non-stationary hydrologic response.

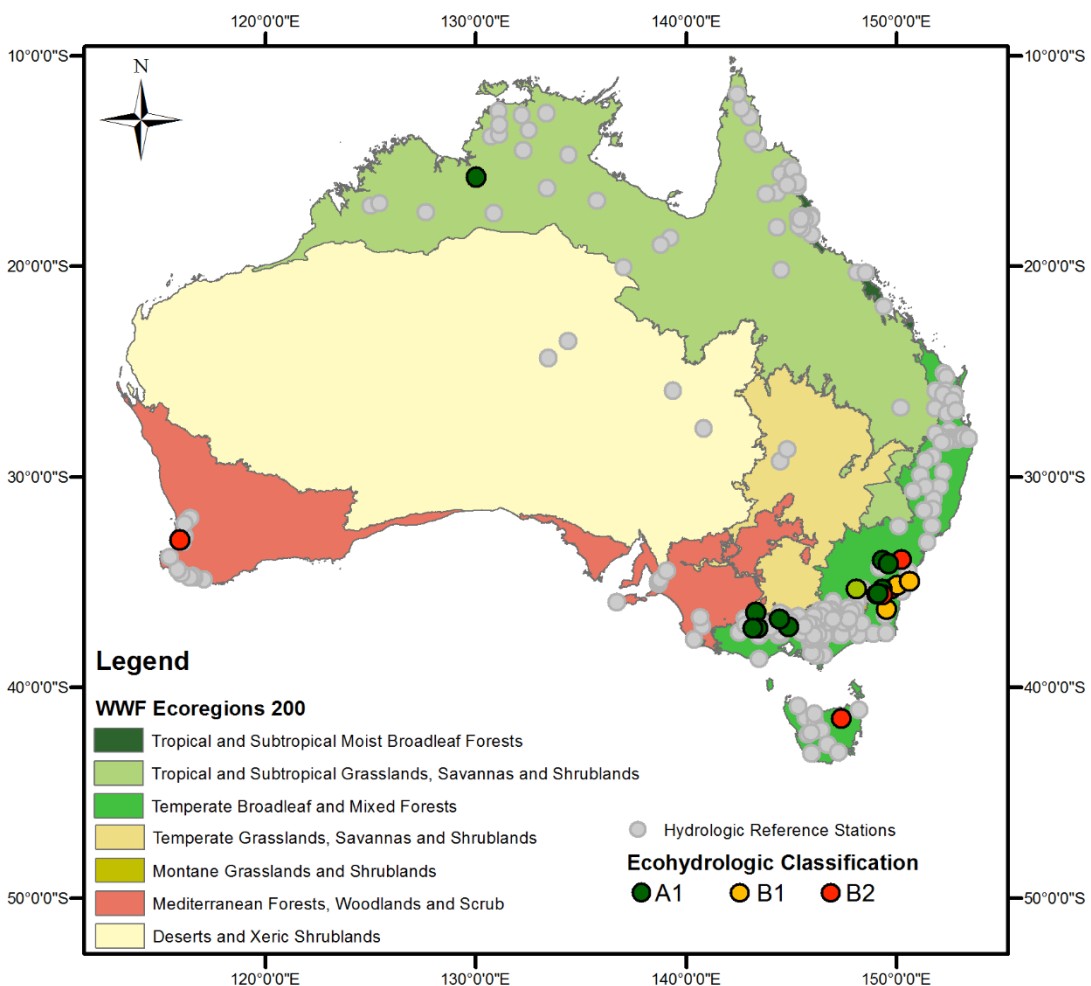

**Figure 1:** Distribution of hydrologic reference stations across Australia. Coloured circles represent catchments with significant trend in annual runoff ratio. Colours represent catchment grouping based on the classification framework of Figure 6. The catchments with non-stationary hydrologic response span over three significant ecoregions of the continent. Ecoregion boundaries are from the World Wildlife Fund (http://maps.tnc.org/gis_data.html).

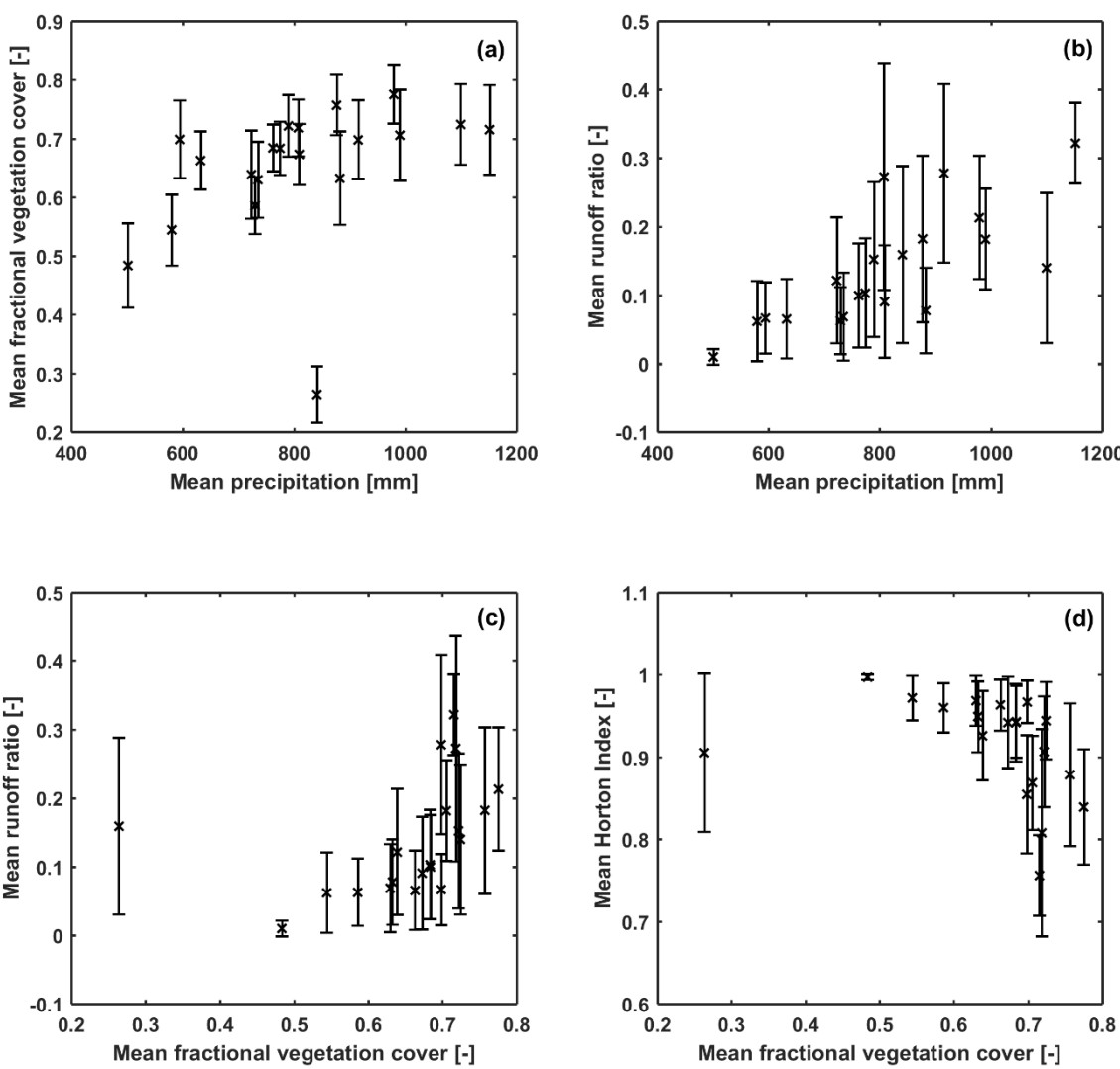

**Figure 2:** Mean and standard deviation of catchment averaged a) annual fractional vegetation cover, and b) annual runoff ratio, against mean annual precipitation (P) in catchments with non-stationary hydrologic response. c) mean and standard deviation of annual runoff ratio, and d) Horton Index versus catchment averaged annual fractional vegetation cover.

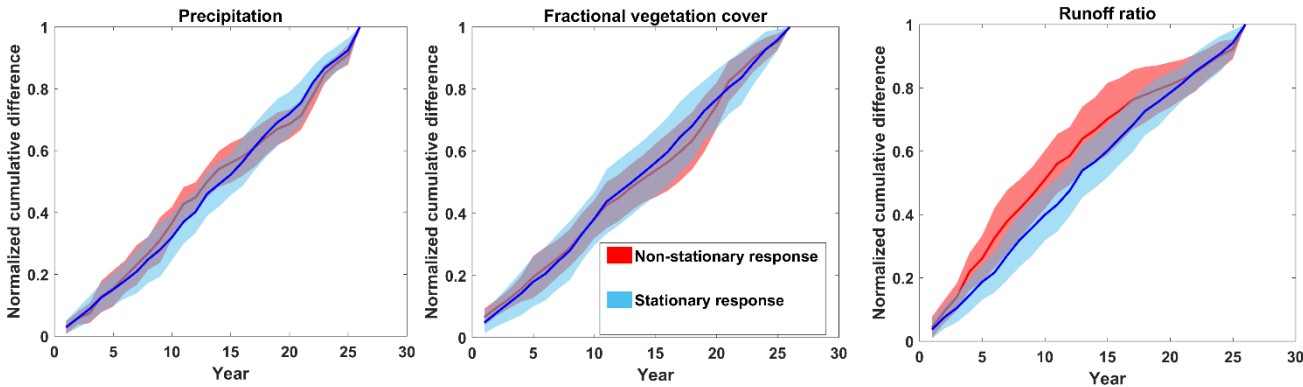

**Figure 3:** Mean normalized cumulative absolute differences in annual precipitation, fractional vegetation cover and runoff ratio between catchments with non-stationary (20 catchments) and stationary (146 catchments) hydrologic response. The shaded areas represent standard deviation.

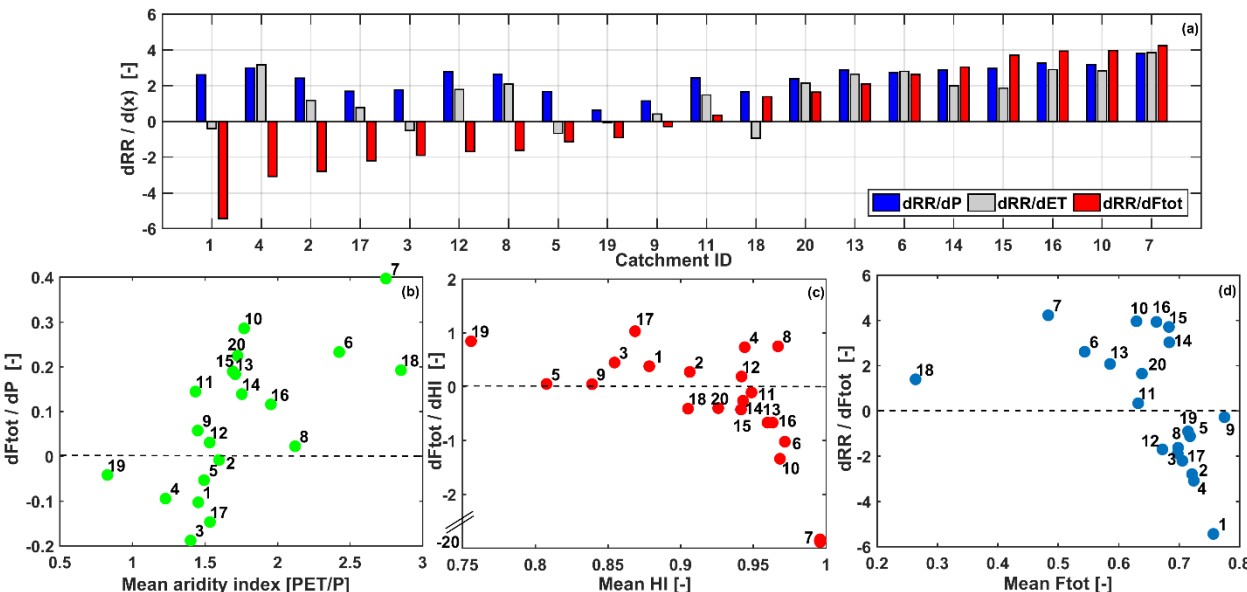

**Figure 4:** a) Normalized sensitivities of runoff ratio (RR) to precipitation (P), water balance ET and fractional vegetation cover (Ftot), normalized sensitivities of annual: b) fractional vegetation cover to precipitation against catchments' mean aridity index, c) fractional vegetation cover to Horton index (HI) against catchments' mean Horton Index, and d) runoff ratio to fractional vegetation cover against catchments' mean fractional vegetation cover in catchments with non-stationary hydrologic response. Data labels refer to the station identification number in Table 1.

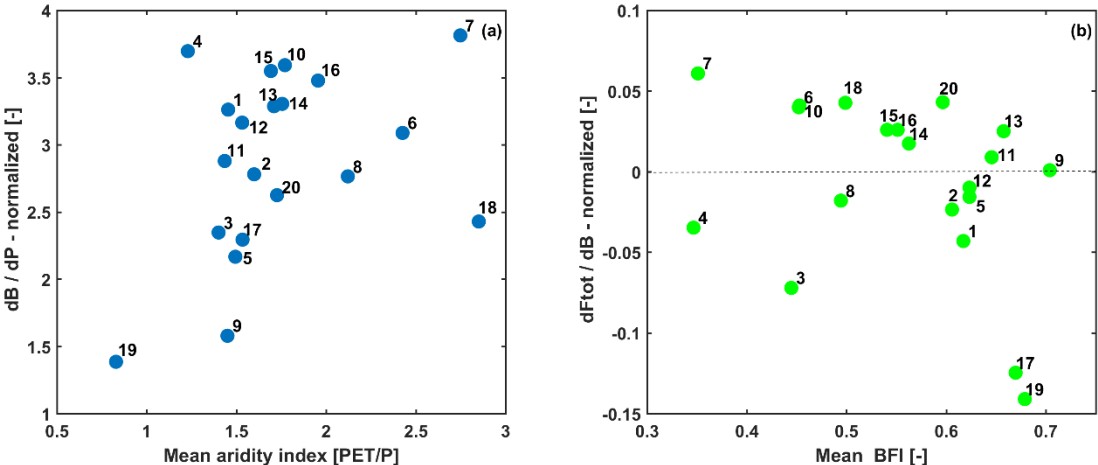

**Figure 5:** a) Normalized baseflow (B) sensitivities to annual precipitation (P) in each catchment with non-stationary hydrologic response against its mean aridity index (1984-2010), b) normalized annual fractional vegetation cover ($F_{tot}$) sensitivities to annual baseflow of each catchment with non-stationary hydrologic response against its mean baseflow index (BFI) (1984-2010). Data labels refer to the station identification number in Table 1. In general, annual baseflow sensitivities to mean annual precipitation decreases in wetter catchments (smaller aridity index). Positive sensitivities of fractional vegetation cover to baseflow decreases in catchments with higher baseflow index. Negative fractional vegetation cover sensitivities to baseflow become more negative in catchments with higher baseflow index indicating larger contribution of groundwater to stream flow.

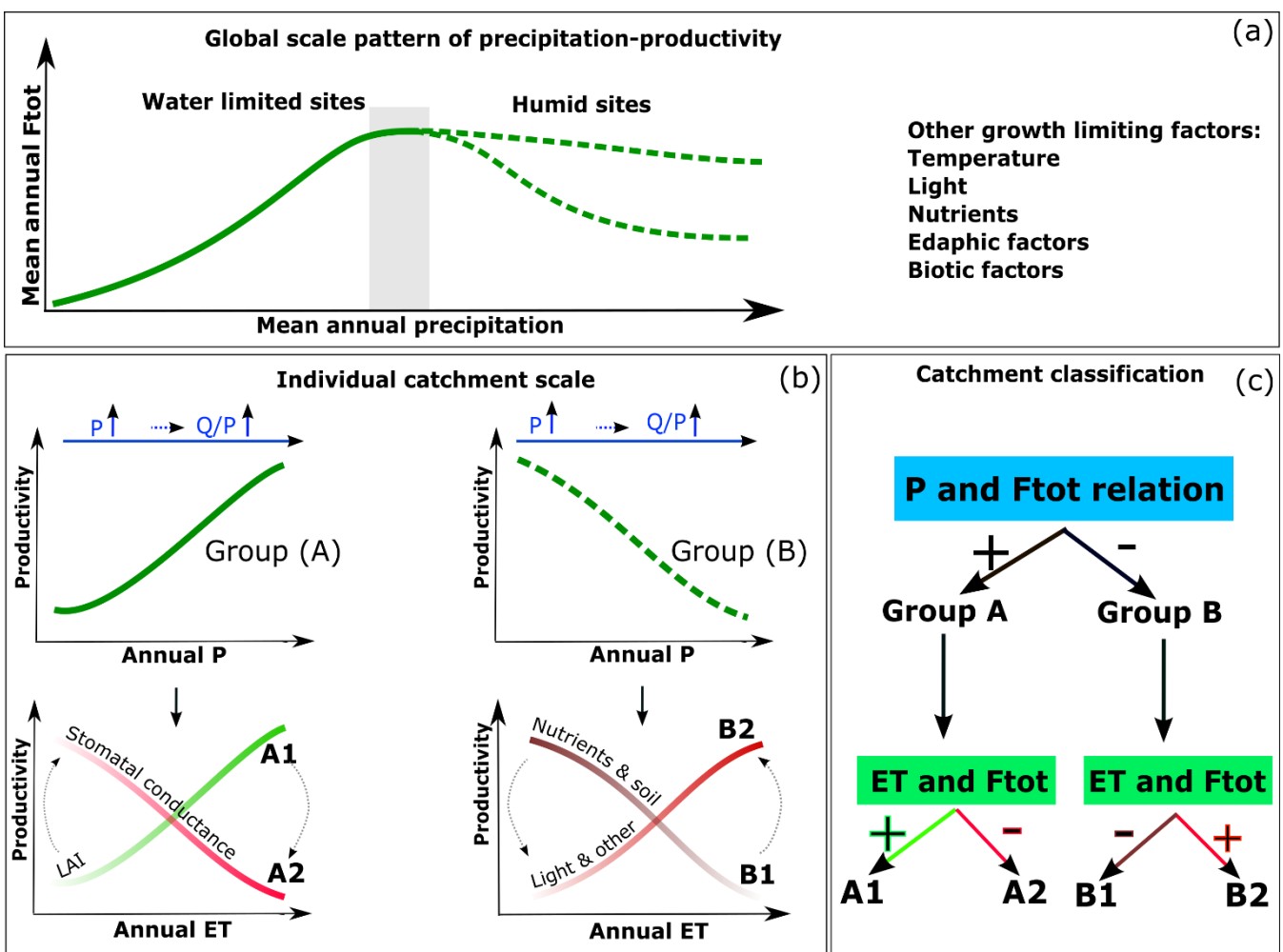

**Figure 6:** a) Global pattern of annual productivity ($F_{tot}$) and mean annual precipitation relationship. While precipitation is the primary factor for vegetation growth in water limited sites, productivity reaches an asymptote in humid areas or decreases (e.g. tropical forests) with increases in precipitation due to biogeochemical or edaphic constraints. The grey region corresponds to catchments in which productivity is insensitive to inter-annual precipitation variability. b) A conceptual framework for characterizing changes in runoff ratio to changes in annual precipitation and vegetation productivity ($F_{tot}$) in relation to catchment's hydroclimatic condition. In group (A) catchments a positive relationship between annual precipitation and productivity exists and annual ET changes in relation to productivity depend on the dominance of structural control (increases in LAI, class A1) versus physiological control (decreases in stomatal conductance, class A2) in controlling productivity. In group (B) catchments, an inverse relationship between precipitation and productivity exists and productivity

is likely constrained by biogeochemical factors. In B1 catchments, negative ET and productivity relation indicates productivity is likely controlled by nutrients availability as drier conditions induce nutrient mineralization. In B2 catchments, light availability and lower temperature reduce ET. In group (A) catchments, runoff ratio would increase as productivity increases while in group (B), runoff ratio will likely decrease with increasing productivity (decreases in precipitation).

5 Depending on the dominance of limiting resource, precipitation-productivity may shift between the two regimes. c) The flowchart illustrates the classification procedure. The classification starts by assessing the correlations between annual precipitation and $F_{tot}$ and then annual ET and $F_{tot}$ in a catchment.

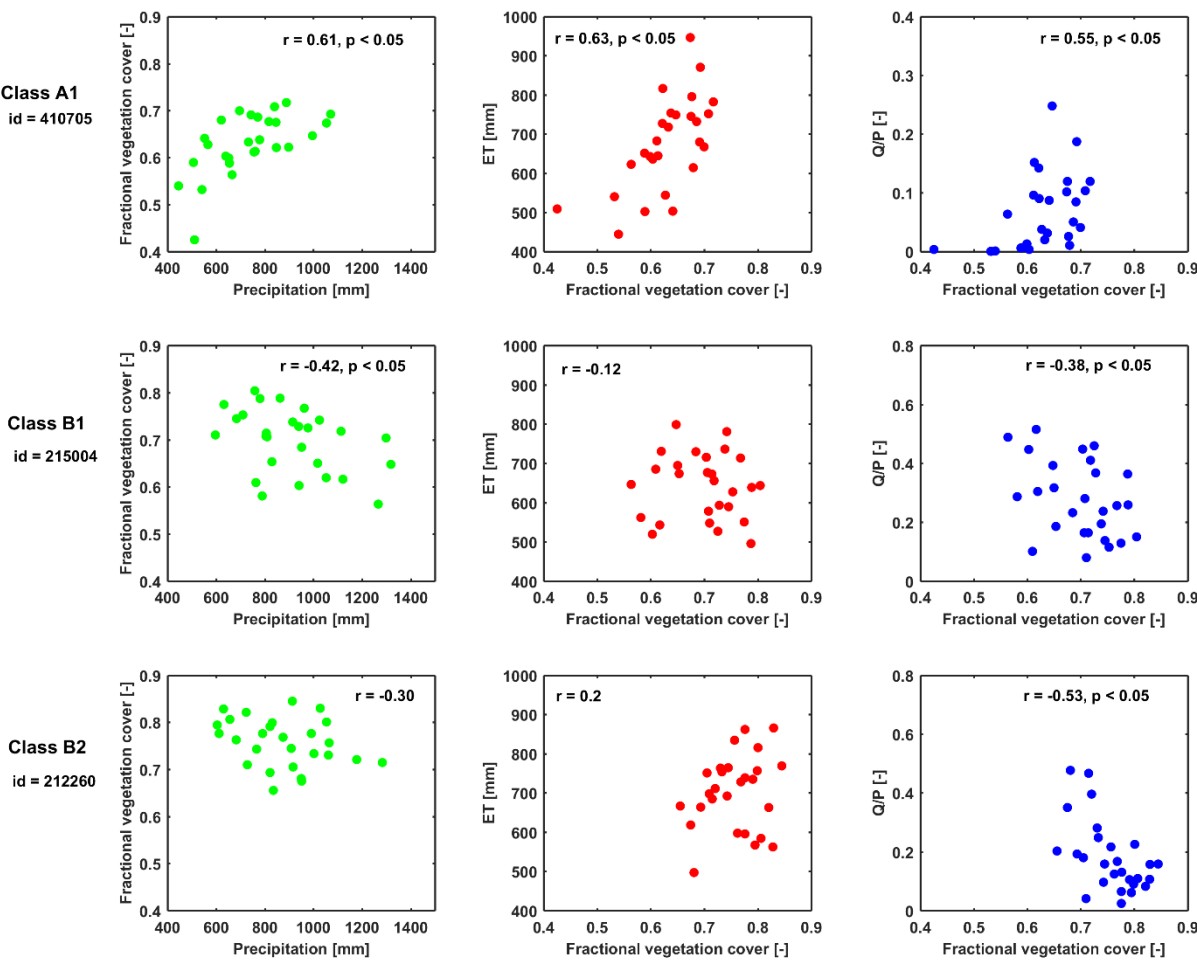

**Figure 7:** Relationships between catchment averaged annual fractional vegetation cover and annual precipitation (left), water balance derived annual ET (middle) and runoff ratio (right) against mean annual fractional vegetation cover (1984-2010) across three catchments representative of each class in Figure 6. The Spearman rank correlation (r) and p-values are shown when correlation is significant.

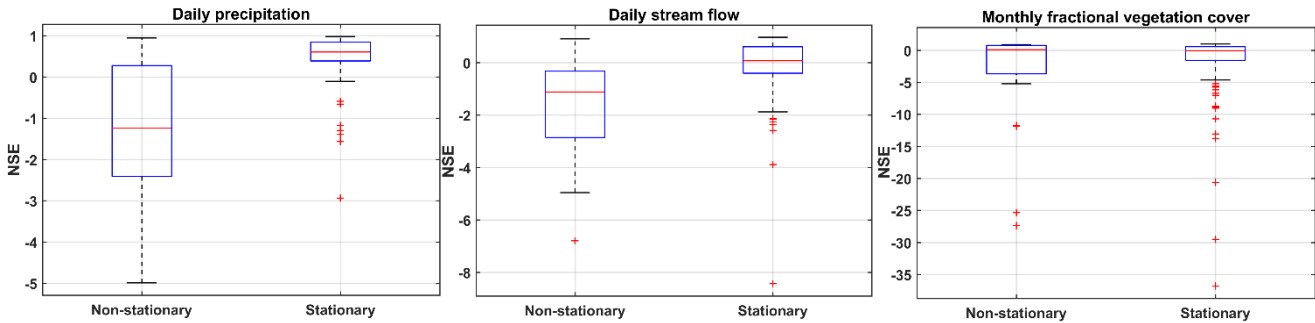

**Figure 8:** Box plots of NSE values calculated between the regime curves of pre-drought and drought periods in catchments with non-stationary (20 catchments) and stationary (146 catchments) hydrologic response respectively. Changes in daily precipitation and runoff and monthly fractional vegetation cover were larger in catchments with non-stationary hydrologic response.