# Peer review of "On the non-stationarity of hydrological response in anthropogenically unaffected catchments: An Australian perspective"

_Hydrology and Earth System Sciences, 2016_

## Referee Comment (RC1) · Anonymous Referee #1 · 20 Aug 2016

This is an interesting study on the ongoing problem of understanding hydrological non-stationarity. I like the work, but I am unclear regarding the robustness of the results as discussed below.

[1] The introduction is well written. I wonder whether there are two other relevant links to be made here. (a) To work on streamflow elasticity (e.g. http://engineering.tufts.edu/cee/people/vogel/documents/climate-elasticty.pdf), and (b) on classification approaches trying to assess nonstationarity (e.g. http://www.hydrol-earth-syst-sci.net/18/273/2014/). I think these two previous approaches might be interesting to connect with here since they both found that a lot of the variability in runoff ratio was difficult to explain and predict.

[2] Similarly, there has been a lot of work on trying to disaggregate the roles of vegetation, storage, energy and moisture on predicting runoff ratio using Budyko type frameworks, which I think also show that it is difficult to come up with simple explanations for reasons for nonstationarity - which I think is line with the results shown here.

[2] In the results section (3.1) the authors state that variables increase, or decrease, or show trends. It would be good if they could quantify these a bit more, rather than just stating that the trends are statistically significant. Especially since the value of such significance tests is regularly questioned (e.g. http://onlinelibrary.wiley.com/doi/10.1002/esp.3618/abstract).

[3] The main question I have relates to the fact that the authors largely focus on analysing the 20 out of 166 catchments for which they saw nonstationarity in the response. While the subsequent analysis of those 20 is fine, I wonder what can be said about the 146 catchment where runoff ratio is not changing? For example, how many of the stationary catchments have experienced precipitation or vegetation or ET changes similar to the ones where runoff ratio changed? That would be a baseline analysis to see whether an interpretation of the causes of runoff ratio nonstationarity are robust. So my main question to the authors is whether they can demonstrate that the catchments not showing runoff ratio change have experienced changes that are smaller regarding the potential driving variables?

---

## Referee Comment (RC2) · Anonymous Referee #2 · 31 Aug 2016

I am very interested in the analysis and discussions about the different influences of vegetation cover and climate changes on runoff in the manuscript. But in my opinion, some analysis is unconvincing and some conclusion is arbitrary. So, I suggest the authors conduct further improvement on the manuscript. Major comments are given below.

1. I suggest that the authors change the usage of "non-stationary catchment". Significant increasing or decreasing doesn't mean that the catchment is not stable. On the contrary, non-significant trend also does not mean stationary. 2. What I am most interested are figure 3 and figure 4. For figure 3b, the authors state that "In catchments with positive precipitation fractional vegetation cover relationships, fractional vegetation cover sensitivities decline with increases in annual precipitation across the catchments". But I would argue that, fractional vegetation cover sensitivity increases significantly with increases in annual precipitation across the catchments when precipitation is smaller than 700 mm; Authors also concluded statement "Fractional vegetation cover sensitivity is highest in the xeric (arid) catchments with lower mean annual precipitation compared to the rest of the non-stationary catchments" from figure 3b. But I cannot see any direct index reflecting "arid". I would suggest that authors plot dFtot/dP against with PET/P in figure 3b, as well as in figure 4a. 3. For figure 3c, the authors should point out: what ranges of HI values mean dry and what HI values mean wet? It is also interesting that in wet regions (low HI), vegetation cover increases when the climate becomes dryer (HI increases)? Authors should give reasonable explanations. 4. For figure 3d, because high Ftot always locates in wet regions. So, according to figure 3d, in dryer regions (low Ftot), runoff coefficient always increases as vegetation cover increases? This is conflict with the conclusion that reforestation and forest growth usually significantly decrease the runoff in dry regions. 5. For figure 4b, the authors concluded that ". . .in catchments where groundwater constitutes significant component of stream flow, fractional vegetation cover exhibits smaller variability. . .". I would also suggest that the authors used the ratio of base flow to total runoff to replace the mean based flow as the x axis. 6. The authors only analyze the vegetation cover besides climate factors. Former studies showed that catchment area and slope etc. are also very important factors, which might significant influences the changes of runoff to climate and vegetation cover changes. The areas of selected catchments ranges from 6.6 to 232846 km2, which might bring unexpected influences on the analysis about figure 3 and 4. That is also probably the reason while only 20/166 catchments showed significant trends in runoff coefficients. So I suggest the authors should consider other catchment factors and explain the underlying reasons. 7. Lack specific data and method descriptions. For example, authors didn't explain how ET and PET were calculated etc.

---

## Author Comment (AC1) · 23 Oct 2016

We thank the reviewers for their valuable and useful comments on this manuscript. We believe that their suggestions will further improve our manuscript and we can address these comments in the revised manuscript. These comments are in line with the complexity of the problem this paper seeks to discuss, and we feel highlights the importance of the paper as a means of adding clarity on how hydrologic models change in the changing world we live in. Please see below our response to each of the reviewers' comment.

**Anonymous Referee #1**

This is an interesting study on the ongoing problem of understanding hydrological nonstationarity. I like the work, but I am unclear regarding the robustness of the results as discussed below.

1. The introduction is well written. I wonder whether there are two other relevant links to be made here. (a) To work on streamflow elasticity (e.g. http://engineering.tufts.edu/cee/people/vogel/documents/climate-elasticty.pdf), and (b) on classification approaches trying to assess nonstationarity (e.g. http://www.hydrolearth-syst-sci.net/18/273/2014/). I think these two previous approaches might be interesting to connect with here since they both found that a lot of the variability in runoff ratio was difficult to explain and predict.

We agree with the reviewer comment to provide a link between the streamflow elasticity approach and the methodology presented here in the revised manuscript. Indeed, normalized sensitivities of runoff ratio to precipitation and fractional vegetation cover in Figure 3a is indicative of elasticity of runoff ratio to changes in precipitation and fractional cover respectively, and this approach is similar to Zheng et al. (2009) for computing climate elasticity of streamflow.

The methodology of Sawicz et al. (2014) to characterize changes in streamflow through catchment classification is interesting. However, the approach requires long term streamflow and climate data records to characterize hydrologic change. While these datasets are available for the Hydrologic Reference Stations in Australia, our methodology is limited by the availability of remotely sensed vegetation products. In the revised Introduction, we will incorporate Sawicz at al. (2014) approach to detect hydrologic change.

2. Similarly, there has been a lot of work on trying to disaggregate the roles of vegetation, storage, energy and moisture on predicting runoff ratio using Budyko type frameworks, which I think also show that it is difficult to come up with simple explanations for reasons for nonstationarity - which I think is line with the results shown here.

We agree with the reviewer comment that it is difficult to disaggregate the role of vegetation, climate and soil moisture on streamflow using the empirical methods such as the Budyko framework or the streamflow elasticity approach. Due to the two-way interactions between catchment water balance and vegetation dynamics, implementation of catchment scale ecohydrologic models is the next logical step to disaggregate the roles of various factors. Nevertheless, previous investigations on assessing climate elasticity of streamflow have shown that the degree of sensitivity of streamflow to various factors depends on the model structure and

calibration approach (Sankarasubramanian et al., 2001). Therefore, further research on both data-based and modeling approaches are required.

3. In the results section (3.1) the authors state that variables increase, or decrease, or show trends. It would be good if they could quantify these a bit more, rather than just stating that the trends are statistically significant. Especially since the value of such significance tests is regularly questioned (e.g. http://onlinelibrary.wiley.com/doi/10.1002/esp.3618/abstract).

We will provide additional information about changes in water balance variables and the rate of trends in the revised manuscript. We agree with the reviewer that the results of the trend analysis are impacted by defining the significance level. While we removed the impact of the start and end year on trend analysis and reduced the impact of autocorrelation on trend analysis, we will present the results of a bootstrap procedure introduced by Douglas et al. (2000) to compute the field significance of regional trend tests in the revised manuscript. In this approach, time series of runoff ratio for every catchment will be resampled 10,000 times using the bootstrap approach. In the next step, the Kendall's S is calculated for each bootstrap sample and regional test statistics is calculated for each iteration. Finally, the CDF of regional test statistics is compared with the historical mean. Our preliminary analysis using the bootstrap approach provided similar results to that presented in the manuscript.

4. The main question I have relates to the fact that the authors largely focus on analysing the 20 out of 166 catchments for which they saw nonstationarity in the response. While the subsequent analysis of those 20 is fine, I wonder what can be said about the 146 catchment where runoff ratio is not changing? For example, how many of the stationary catchments have experienced precipitation or vegetation or ET changes similar to the ones where runoff ratio changed? That would be a baseline analysis to see whether an interpretation of the causes of runoff ratio nonstationarity are robust. So my main question to the authors is whether they can demonstrate that the catchments not showing runoff ratio change have experienced changes that are smaller regarding the potential driving variables?

We agree with the reviewer comment to provide a baseline analysis to show whether stationary catchments experienced similar changes in precipitation, runoff and vegetation compared to the catchments with non-stationary hydrologic response. To show these differences, we will implement the approach of Coopersmith et al. (2014) by developing regime curves based on daily runoff, precipitation and monthly fractional vegetation cover for each catchment using pre-drought and drought period data. Our preliminary analysis shows that in some cases, large changes in the regime curves have been observed particularly in catchments with non-stationary response.

**Anonymous Referee #2**

I am very interested in the analysis and discussions about the different influences of vegetation cover and climate changes on runoff in the manuscript. But in my opinion, some analysis is unconvincing and some conclusion is arbitrary. So, I suggest the authors conduct further improvement on the manuscript. Major comments are given below.

1. I suggest that the authors change the usage of "non-stationary catchment". Significant increasing or decreasing doesn't mean that the catchment is not stable. On the contrary, non-significant trend also does not mean stationary.

Thank you for providing this comment. The term "non-stationary catchment" is used for a matter of brevity in the manuscript. In some cases, we have used the term "catchments with non-stationary hydrologic response" in the manuscript. We will clarify the above usage further in the revised manuscript.

2. What I am most interested are figure 3 and figure 4. For figure 3b, the authors state that "In catchments with positive precipitation fractional vegetation cover relationships, fractional vegetation cover sensitivities decline with increases in annual precipitation across the catchments". But I would argue that, fractional vegetation cover sensitivity increases significantly with increases in annual precipitation across the catchments when precipitation is smaller than 700 mm; Authors also concluded statement "Fractional vegetation cover sensitivity is highest in the xeric (arid) catchments with lower mean annual precipitation compared to the rest of the non-stationary catchments" from figure 3b. But I cannot see any direct index reflecting "arid". I would suggest that authors plot dFtot/dP against with PET/P in figure 3b, as well as in figure 4a.

We will revise this statement in the revised manuscript to state that "across catchments with positive precipitation fractional vegetation cover relationships, fractional vegetation coverage sensitivity approaches zero in catchments with higher mean annual precipitation".

We will incorporate reviewer comment to show aridity-index with dFtot/dP in the revised manuscript and similarly for figure 4a. Catchment 7 with the lowest amount of mean annual precipitation has the largest aridity-index among non-stationary catchments.

3. For figure 3c, the authors should point out: what ranges of HI values mean dry and what HI values mean wet? It is also interesting that in wet regions (low HI), vegetation cover increases when the climate becomes dryer (HI increases)? Authors should give reasonable explanations.

In arid and semiarid catchments, quick flow constitutes most of the total streamflow (S is almost equal to total runoff in equation 3). Therefore, we expect HI to approach 1 in arid catchments. In humid catchments, quick flow runoff is smaller than the total stream flow and HI is less than 1. In catchments with limited storage, HI is undefined (0/0) (Troch et al., 2009). We will clarify these ranges in the revised manuscript. Please see Troch et al. (2009) for additional details.

The second question is a very important point and it is a subject of further investigations to identify the exact cause of vegetation increase under dryer conditions in group B catchments. One plausible mechanism as discussed here and in an earlier paper by Brooks et al. (2011) is nutrient limitation

as similar behavior is also observed in some of the MOPEX catchments located in the humid climate. In this paper, we hypothesize that nutrient, light and temperature limitations may contribute to the observed response. With limited data on sunshine hours, we were able to show that in some of these catchments light limitation contributes to the observed pattern. However, no information about nutrient content is available to test this hypothesis. We also used various remote sensing datasets to make sure the observed pattern is not the artifact of remote sensing data. The next step is to use ecohydrologic models that can incorporate nutrient limitation in simulating carbon dynamics and vegetation growth.

4. For figure 3d, because high Ftot always locates in wet regions. So, according to figure 3d, in dryer regions (low Ftot), runoff coefficient always increases as vegetation cover increases? This is conflict with the conclusion that reforestation and forest growth usually significantly decrease the runoff in dry regions.

To clarify this point, we refer to Figure 5b where interactions between precipitation-fractional vegetation cover and runoff ratio are outlined. As can be seen in Figure 5b, positive correlations between precipitation and fractional vegetation cover exist in water limited catchments (Group A). This means that higher precipitation increases productivity and Q/P. As can be seen in Figure 3a and 3b, sensitivity of runoff ratio to fractional vegetation cover is positive in drier catchments (water limited catchments based on our classification). As period of higher productivity coincides with higher precipitation (positive precipitation-fractional vegetation cover relationship) in these catchments, runoff ratio increases in years with higher precipitation. It should be noted that the percentage of tree cover in these drier catchments are more than 60% with a few exceptions (Table S1, supplementary Information). In Group B catchments percent tree cover is higher than water limited catchments. Overall, mean annual runoff ratio and its variability (standard deviation) are smaller in drier catchments with smaller mean fractional vegetation cover (Figure 2). We will clarify this point in the revised manuscript.

5. For figure 4b, the authors concluded that ": in catchments where groundwater constitutes significant component of stream flow, fractional vegetation cover exhibits smaller variability: : :". I would also suggest that the authors used the ratio of base flow to total runoff to replace the mean based flow as the x axis.

We will use baseflow index instead of mean baseflow in the revised manuscript.

6. The authors only analyze the vegetation cover besides climate factors. Former studies showed that catchment area and slope etc. are also very important factors, which might significant influences the changes of runoff to climate and vegetation cover changes. The areas of selected catchments ranges from 6.6 to 232846 $km^2$, which might bring unexpected influences on the analysis about figure 3 and 4. That is also probably the reason while only 20/166 catchments showed significant trends in runoff coefficients. So I suggest the authors should consider other catchment factors and explain the underlying reasons.

We will explore the impact of slope and area in the revised manuscript. However, within non-stationary catchments no significant differences in catchment mean slope exist and catchment area ranges from 18.7 to 5158 $km^2$ (Table 1).

7. Lack specific data and method descriptions. For example, authors didn't explain how ET and PET were calculated etc.

Thank you for your comment. We will include detailed descriptions of ET and PET computations in the revised manuscript.

**References:**

Coopersmith, E. J., B. S. Minsker, and M. Sivapalan (2014), Patterns of regional hydroclimatic shifts: An analysis of changing hydrologic regimes, *Water Resour. Res.*, *50*(3), 1960–1983, doi:10.1002/2012WR013320.

Douglas, E. M., R. M. Vogel, and C. N. Kroll (2000), Trends in floods and low flows in the United States: impact of spatial correlation, *J. Hydrol.*, *240*(1–2), 90–105, doi:http://dx.doi.org/10.1016/S0022-1694(00)00336-X.

Sankarasubramanian, A., R. M. Vogel, and J. F. Limbrunner (2001), Climate elasticity of streamflow in the United States, *Water Resour. Res.*, *37*(6), 1771–1781, doi:10.1029/2000WR900330.

Sawicz, K. A., C. Kelleher, T. Wagener, P. Troch, M. Sivapalan, and G. Carrillo (2014), Characterizing hydrologic change through catchment classification, *Hydrol. Earth Syst. Sci.*, *18*(1), 273–285, doi:10.5194/hess-18-273-2014.

Zheng, H., L. Zhang, R. Zhu, C. Liu, Y. Sato, and Y. Fukushima (2009), Responses of streamflow to climate and land surface change in the headwaters of the Yellow River Basin, *Water Resour. Res.*, *45*(7), doi:10.1029/2007WR006665.

---

## Author Response (AR2)

We thank the reviewers for their valuable and useful comments on this manuscript. We believe that their suggestions will further improve our manuscript and we can address these comments in the revised manuscript. These comments are in line with the complexity of the problem this paper seeks to discuss, and we feel highlights the importance of the paper as a means of adding clarity on how hydrologic models change in the changing world we live in.

Major changes in the revised manuscript are:

1) Adding two additional figures (Figure 3 and 8) to illustrate differences between catchments that exhibit stationary or non-stationary behavior.
2) Adding section 4.1 to assess changes in vegetation and water balance variables regime curves between catchments that exhibit stationary or non-stationary behavior.
3) Determining the field significance level for regional trend analysis.
4) Assessing the impact of catchment slope on non-stationary response.

Please see below our response to each of the reviewers' comment.

**Anonymous Referee #1**

This is an interesting study on the ongoing problem of understanding hydrological nonstationarity. I like the work, but I am unclear regarding the robustness of the results as discussed below.

1. The introduction is well written. I wonder whether there are two other relevant links to be made here. (a) To work on streamflow elasticity (e.g. http://engineering.tufts.edu/cee/people/vogel/documents/climate-elasticty.pdf), and (b) on classification approaches trying to assess nonstationarity (e.g. http://www.hydroearth-syst-sci.net/18/273/2014/). I think these two previous approaches might be interesting to connect with here since they both found that a lot of the variability in runoff ratio was difficult to explain and predict.

We agree with the reviewer comment to provide a link between the streamflow elasticity approach and the methodology presented here in the revised manuscript. Indeed, normalized sensitivities of runoff ratio to precipitation and fractional vegetation cover in Figure 3a is indicative of elasticity of runoff ratio to changes in precipitation and fractional cover respectively, and this approach is similar to Zheng et al. (2009) for computing climate elasticity of streamflow.

The methodology of Sawicz et al. (2014) to characterize changes in streamflow through catchment classification is interesting. However, the approach requires long term streamflow and climate data records to characterize hydrologic change. While these datasets are available for the Hydrologic Reference Stations in Australia, our methodology is limited by the availability of remotely sensed vegetation products. In the revised Introduction, we will incorporate Sawicz at al. (2014) approach to detect hydrologic change.

To incorporate the reviewer comment, the Introduction and Methods sections are revised as follows:

Introduction: Sawicz et al. (2014) illustrated that changes in climate characteristics of catchments can mostly explain hydrologic change which was characterized by changes in groupings of 314 catchments in the USA. Due to the lack of information, temporal changes in land use were not considered in characterizing hydrologic change in this approach.

Section 2.2.3.: Normalized sensitivity of annual runoff ratio is equivalent to the stream flow elasticity approach of Zheng et al. (2009) that defined stream flow elasticity as the linear regression coefficient between the proportional changes in streamflow and a climatic variable (precipitation or potential evapotranspiration).

2. Similarly, there has been a lot of work on trying to disaggregate the roles of vegetation, storage, energy and moisture on predicting runoff ratio using Budyko type frameworks, which I think also show that it is difficult to come up with simple explanations for reasons for nonstationarity - which I think is line with the results shown here.

We agree with the reviewer comment that it is difficult to disaggregate the role of vegetation, climate and soil moisture on streamflow using the empirical methods such as the Budyko framework or the streamflow elasticity approach. Due to the two-way interactions between catchment water balance and vegetation dynamics, implementation of catchment scale ecohydrologic models is the next logical step to disaggregate the roles of various factors. Nevertheless, previous investigations on assessing climate elasticity of streamflow have shown that the degree of sensitivity of streamflow to various factors depends on the model structure and calibration approach (Sankarasubramanian et al., 2001). Therefore, further research on both data-based and modeling approaches are required.

To incorporate the reviewer comment, the following sentence is added to the Introduction:

Similarly, assessing climate elasticity of stream flow has shown that the degree of sensitivity of stream flow to various factors depends on the model structure and calibration approach (Sankarasubramanian et al., 2001).

3. In the results section (3.1) the authors state that variables increase, or decrease, or show trends. It would be good if they could quantify these a bit more, rather than just stating that the trends are statistically significant. Especially since the value of such significance tests is regularly questioned (e.g. http://onlinelibrary.wiley.com/doi/10.1002/esp.3618/abstract).

We will provide additional information about changes in water balance variables and the rate of trends in the revised manuscript. The values of linear trend for precipitation, runoff ratio and fractional vegetation cover are also shown in Table 1.

Results 3.1.: An increasing trend for runoff ratio in the East Baines River (0.009/yr) is consistent with annual precipitation increases (13.2 mm/yr). Moreover, this catchment has the smallest fractional vegetation cover (0.26) amongst the non-stationary catchments. The North Esk catchment in Tasmania is the only catchment amongst the catchments with non-stationary response in which runoff ratio declined despite increases in annual precipitation (6.4 mm/yr)

(Table 1). In the Tasmanian catchment, the increasing trend in fractional vegetation cover (0.009/yr) is significant and results in ET increase and subsequently lower runoff ratio during 1984-2005 period. In the rest of the non-stationary catchments, total annual precipitation decreased between -1.9 mm/yr to -24.7 mm/yr in 1984-2005 period which is consistent with the decreasing trend in annual runoff ratio (-0.0008/yr to -0.016/yr).

To summarize changes in precipitation, fractional vegetation cover and runoff ratio for each catchment, the cumulative absolute differences between consecutive annual values of each variable are calculated and normalized by the total absolute difference. To illustrate differences between catchments with non-stationary and stationary hydrologic response, the mean and standard deviations of normalized cumulative differences are calculated for each group. As it can be seen in Figure 3, normalized cumulative differences in annual precipitation and fractional vegetation cover between the catchments with non-stationary and stationary hydrologic response are very similar. However, large differences in the normalized cumulative differences of annual runoff ratio exist between the stationary and non-stationary catchments.

To incorporate reviewer comment, we added the following paragraph to section 3.2:

To explore differences between catchments with non-stationary or stationary behaviour, the cumulative absolute differences between consecutive annual values of precipitation, fractional vegetation cover and runoff ratio for each catchment are calculated and normalized by the total absolute difference. In Figure 3, the differences between catchments with non-stationary and stationary hydrologic response are illustrated by presenting the mean and standard deviations of normalized cumulative differences for each group. As can be seen in Figure 3, normalized cumulative differences in annual precipitation and fractional vegetation cover between the catchments with non-stationary and stationary hydrologic response are very similar. However, large differences in the normalized cumulative differences of annual runoff ratio exist between these catchments. The catchment area ranges from 18.7 to 5158 km$^2$ in catchments with non-stationary hydrologic response (Table 1). While increases in runoff ratio, P-Q and mean fractional vegetation cover with increases in mean catchment slope are observed in catchments with non-stationary hydrologic response, no distinct differences between catchments with stationary and non-stationary hydrologic response are observed.

[Figure]

**Figure 3:** Mean normalized cumulative absolute differences in annual precipitation, fractional vegetation cover and runoff ratio between catchments with non-stationary (20 catchments) and stationary (146 catchments) hydrologic response. The shaded areas represent standard deviation.

We agree with the reviewer that the results of the trend analysis are impacted by defining the significance level. While we removed the impact of the start and end year on trend analysis and reduced the impact of autocorrelation on trend analysis, we will present the results of a bootstrap procedure introduced by Douglas et al. (2000) to compute the field significance of regional trend tests in the revised manuscript. In this approach, time series of runoff ratio for every catchment are resampled 10,000 times using the bootstrap approach. In the next step, the Kendall's S is calculated for each bootstrap sample and the regional test statistics is calculated by averaging Kendall's S for each iteration and computing non-exceedance probability using the Weibull plotting position formula. Finally, the CDF of regional test statistics is compared with the historical Kendall's S calculated for each station using 0.01 significance level. Our preliminary analysis using the bootstrap approach provided similar results to that presented in the manuscript. Indeed, the field significance level obtained from the bootstrap samples is 0.0239 which is more relaxed than the p-value = 0.01 originally used in the manuscript. Using the new field significance level, 34 catchments will be classified as non-stationary.

To address this comment in the manuscript, please see the response to comment 4.

[Figure]

4. The main question I have relates to the fact that the authors largely focus on analysing the 20 out of 166 catchments for which they saw nonstationarity in the response. While the subsequent analysis of those 20 is fine, I wonder what can be said about the 146 catchment where runoff ratio is not changing? For example, how many of the stationary catchments have experienced precipitation or vegetation or ET changes similar to the ones where runoff ratio changed? That would be a baseline analysis to see whether an interpretation of the causes of runoff ratio nonstationarity are robust. So my main question to the authors is whether they can demonstrate that the catchments not showing runoff ratio change have experienced changes that are smaller regarding the potential driving variables?

We agree with the reviewer comment to provide a baseline analysis to show whether stationary catchments experienced similar changes in precipitation, runoff and vegetation compared to the catchments with non-stationary hydrologic response. To show these differences, we implemented the approach of Coopersmith et al. (2014) by developing regime curves based on daily runoff, precipitation and monthly fractional vegetation cover data for each catchment using pre-drought (1984-1996) and drought period (1997-2009) data. Our analysis shows that in some cases, large changes in the regime curves have been observed particularly in catchments with

non-stationary hydrologic response. To summarize the differences between the regime curves for pre-drought and drought periods, Nash Sutcliffe Efficiency (NSE) criterion is calculated. As can be seen in Figure 8, differences in daily precipitation and runoff and monthly fractional vegetation cover regime curves are much higher (indicated by negative NSE) in catchments with non-stationary hydrologic response than the catchments that do not exhibit non-stationary behavior.

To address this comment, the following section is added to the Discussion section:

**4.1. Did catchments with non-stationary hydrologic response experience similar changes in vegetation and water balance variables?**

To explore whether HRS catchments are undergone similar changes during the period of analysis, regime curves based on daily runoff, precipitation and monthly fractional vegetation cover data for each catchment are developed using data from pre-drought (1984-1996) and drought period (1997-2009) (Coopersmith et al., 2014). Regime curves are obtained by averaging daily values of precipitation or runoff for a given day over the length of the data. As daily fractional vegetation cover data are not available, monthly values are used to develop the regime curves. To summarize the differences between the regime curves for the pre-drought and drought periods, Nash Sutcliffe Efficiency (NSE) criterion is calculated. As can be seen in Figure 8, differences in daily precipitation and runoff and monthly fractional vegetation cover regime curves are much higher (indicated by negative NSE) in catchments with non-stationary hydrologic response than the catchments that do not exhibit non-stationary behavior. While the results of trend analysis are impacted by defining the significance level, the above analysis indicates that catchments with non-stationary behaviour have undergone larger changes. To further assess the impact of significance level on the results of the trend analysis, the approach of Douglas et al. (2000) for computing the field significance of regional trend tests are implemented. In this approach, time series of runoff ratio for every catchment are resampled 10,000 times using the bootstrap approach. In the next step, the Kendall's S is calculated for each bootstrap sample and the regional test statistics is calculated by averaging Kendall's S for each iteration and computing non-exceedance probability using the Weibull plotting position formula. Finally, the CDF of regional test statistics is compared with the historical Kendall's S calculated for each station using 0.01 significance level. Indeed, the field significance level obtained from the bootstrap samples is 0.0239 which is more relaxed than the p-value = 0.01 originally used. Using the new field significance level, 34 catchments are classified as non-stationary.

[Figure]

**Figure 8:** Box plots of NSE values calculated between the regime curves of pre-drought and drought periods in catchments with non-stationary (20 catchments) and stationary (146 catchments) hydrologic response respectively. Changes in daily precipitation and runoff and monthly fractional vegetation cover were larger in catchments with non-stationary hydrologic response.

**Anonymous Referee #2**

I am very interested in the analysis and discussions about the different influences of vegetation cover and climate changes on runoff in the manuscript. But in my opinion, some analysis is unconvincing and some conclusion is arbitrary. So, I suggest the authors conduct further improvement on the manuscript. Major comments are given below.

1. I suggest that the authors change the usage of "non-stationary catchment". Significant increasing or decreasing doesn't mean that the catchment is not stable. On the contrary, non-significant trend also does not mean stationary.

Thank you for providing this comment. The term "non-stationary catchment" is used for a matter of brevity in the manuscript. In some cases, we have used the term "catchments with non-stationary hydrologic response" in the manuscript. We clarified the above usage further in the revised manuscript.

For example, the following sentence in the abstract is revised as follows:
Runoff ratio decreased across the catchments with non-stationary hydrologic response with the exception of one catchment in northern Australia.

2. What I am most interested are figure 3 and figure 4. For figure 3b, the authors state that "In catchments with positive precipitation fractional vegetation cover relationships, fractional vegetation cover sensitivities decline with increases in annual precipitation across the catchments". But I would argue that, fractional vegetation cover sensitivity increases significantly with increases in annual precipitation across the catchments when precipitation is smaller than 700 mm; Authors also concluded statement "Fractional vegetation cover sensitivity is highest in the xeric (arid) catchments with lower mean annual precipitation compared to the rest of the non-stationary catchments" from figure 3b. But I cannot see any direct index

reflecting "arid". I would suggest that authors plot dFtot/dP against with PET/P in figure 3b, as well as in figure 4a.

The purpose of this sentence is to state that "across catchments with positive precipitation fractional vegetation cover relationships, fractional vegetation coverage sensitivity approaches zero in catchments with higher mean annual precipitation". However, as we replaced mean annual precipitation with the mean aridity index based on the reviewer comment, this statement is revised as follows:

Section 3.3.: across catchments with positive precipitation-fractional vegetation cover relationships, fractional vegetation cover sensitivities approach zero in catchments with aridity index of 1.5.

We incorporated the reviewer comment to show aridity-index with dFtot/dP in the revised manuscript and similarly for figure 5a. Catchment 18 and 7 have the largest aridity-index among catchments with non-stationary hydrologic response and are classified as semi-arid based on the UNEP classification.

In the revised manuscript, we replaced xeric with "semi-arid".
Section 3.3: Fractional vegetation cover sensitivity is highest in the semi-arid catchments with lower mean annual precipitation compared to the rest of the catchments with non-stationary hydrologic response. In catchments with mean annual precipitation of 800 mm or higher (aridity index < 1.5), slopes of fractional vegetation cover to mean annual precipitation are zero or negative.

[Figure]

**Figure 4:** a) Normalized sensitivities of runoff ratio (RR) to precipitation (P), water balance ET and fractional vegetation cover (Ftot), normalized sensitivities of annual: b) fractional vegetation cover to precipitation against catchments' mean aridity index, c) fractional vegetation cover to Horton index (HI) against catchments' mean Horton Index, and d) runoff ratio to fractional vegetation cover against catchments' mean fractional vegetation cover in catchments with non-stationary hydrologic response. Data labels refer to the station identification number in Table 1.

[Figure]

**Figure 5.** a) Normalized baseflow (B) sensitivities to annual precipitation (P) in each catchment with non-stationary hydrologic response against its mean aridity index (1984-2010), b) normalized annual fractional vegetation cover ($F_{tot}$) sensitivities to annual baseflow of each catchment with non-stationary hydrologic response against its mean baseflow index (BFI) (1984-2010). Data labels refer to the station identification number in Table 1. In general, annual baseflow sensitivities to mean annual precipitation decreases in wetter catchments (smaller aridity index). Positive sensitivities of fractional vegetation cover to baseflow decreases in catchments with higher baseflow index. Negative fractional vegetation cover sensitivities to baseflow become more negative in catchments with higher baseflow index indicating larger contribution of groundwater to stream flow.

3. For figure 3c, the authors should point out: what ranges of HI values mean dry and what HI values mean wet? It is also interesting that in wet regions (low HI), vegetation cover increases when the climate becomes dryer (HI increases)? Authors should give reasonable explanations.

In arid and semiarid catchments, quick flow constitutes most of the total streamflow (S is almost equal to total runoff in equation 3). Therefore, we expect HI to approach 1 in arid catchments. In humid catchments, quick flow runoff is smaller than the total stream flow and HI is less than 1. In catchments with limited storage, HI is undefined (0/0) (Troch et al., 2009). We will clarify these ranges in the revised manuscript. Please see Troch et al. (2009) for additional details.

In section 2.2.2 we added the following sentences to incorporate reviewer comment:
In arid and semiarid catchments as quick flow constitutes most of the total stream flow, HI is approaching 1. In humid catchments, quick flow runoff is smaller than the total stream flow and HI is less than 1. In catchments with limited storage, HI is undefined (0/0) (Troch et al., 2009).

The second question is a very important point and it is a subject of further investigations to identify the exact cause of vegetation increase under dryer conditions in group B catchments. One plausible mechanism as discussed here and in an earlier paper by Brooks et al. (2011) is nutrient limitation as similar behavior is also observed in some of the MOPEX catchments located in the humid climate. In this paper, we hypothesize that nutrient, light and temperature limitations may contribute to the observed response. With limited data on sunshine hours, we were able to show that in some of these catchments light limitation contributes to the observed pattern. However, no information about nutrient content is available to test this hypothesis. We

also used various remote sensing datasets to make sure the observed pattern is not the artifact of remote sensing data. The next step is to use ecohydrologic models that can incorporate nutrient limitation in simulating carbon dynamics and vegetation growth.

4. For figure 3d, because high Ftot always locates in wet regions. So, according to figure 3d, in dryer regions (low Ftot), runoff coefficient always increases as vegetation cover increases? This is conflict with the conclusion that reforestation and forest growth usually significantly decrease the runoff in dry regions.

To clarify this point, we refer to Figure 6b where interactions between precipitation-fractional vegetation cover and runoff ratio are outlined. As can be seen in Figure 6b, positive correlations between precipitation and fractional vegetation cover exist in water limited catchments (Group A). This means that higher precipitation increases productivity and Q/P. As can be seen in Figure 4a and 4b of the revised manuscript, sensitivity of runoff ratio to fractional vegetation cover is positive in drier catchments (water limited catchments based on our classification). As period of higher productivity coincides with higher precipitation (positive precipitation-fractional vegetation cover relationship) in these catchments, runoff ratio increases in years with higher precipitation. It should be noted that the percentage of tree cover in these drier catchments are more than 60% with a few exceptions (Table S1, supplementary Information). In Group B catchments percent tree cover is higher than water limited catchments. Overall, mean annual runoff ratio and its variability (standard deviation) are smaller in drier catchments with smaller mean fractional vegetation cover (Figure 2). We clarified this point in the revised manuscript.

Section 3.3: Across water limited catchments (positive runoff ratio-fractional vegetation cover relationship), runoff ratio's sensitivities are smallest in catchments with the highest vegetation cover. As periods of higher productivity coincide with higher precipitation (positive precipitation-fractional vegetation cover relationship) in these catchments, runoff ratio increases in years with higher precipitation. It should be noted that the percentage of tree cover in these drier catchments are more than 60% with a few exceptions (Table S1, supplementary Information). Negative runoff ratio-fractional vegetation cover sensitivities become more negative in catchments with higher fractional vegetation cover. Overall, mean annual runoff ratio and its variability (standard deviation) are smaller in drier catchments with smaller mean fractional vegetation cover (Fig. 2).

5. For figure 4b, the authors concluded that ": in catchments where groundwater constitutes significant component of stream flow, fractional vegetation cover exhibits smaller variability: : :". I would also suggest that the authors used the ratio of base flow to total runoff to replace the mean based flow as the x axis.

We used baseflow index instead of mean baseflow in Figure 5b in the revised manuscript.

[Figure]

**Figure 4.** a) Normalized baseflow (B) sensitivities to annual precipitation (P) in each catchment with non-stationary hydrologic response against its mean aridity index (1984-2010), b) normalized annual fractional vegetation cover ($F_{tot}$) sensitivities to annual baseflow of each catchment with non-stationary hydrologic response against its mean baseflow index (BFI) (1984-2010). Data labels refer to the station identification number in Table 1. In general, annual baseflow sensitivities to mean annual precipitation decreases in wetter catchments (smaller aridity index). Positive sensitivities of fractional vegetation cover to baseflow decreases in catchments with higher baseflow index. Negative fractional vegetation cover sensitivities to baseflow become more negative in catchments with higher baseflow index indicating larger contribution of groundwater to stream flow.

Section 3.3: …We used baseflow as a measure of catchment storage response to inter-annual precipitation variability. Baseflow's sensitivities to mean annual aridity index are highest in drier catchments with non-stationary hydrologic response (Fig. 5a). Normalized fractional vegetation cover sensitivities to the baseflow decrease in catchments with higher annual baseflow index and even become negative at higher baseflow indices (Fig. 5b). This result suggests that in catchments where groundwater constitutes significant component of stream flow, fractional vegetation cover exhibits smaller variability to changes in baseflow as vegetation roots have access to deeper water storage for transpiration and have less sensitivity to changes in baseflow.

6. The authors only analyze the vegetation cover besides climate factors. Former studies showed that catchment area and slope etc. are also very important factors, which might significant influences the changes of runoff to climate and vegetation cover changes. The areas of selected catchments ranges from 6.6 to 232846 km², which might bring unexpected influences on the analysis about figure 3 and 4. That is also probably the reason while only 20/166 catchments showed significant trends in runoff coefficients. So I suggest the authors should consider other catchment factors and explain the underlying reasons.

We explored the impact of slope and area to further investigate observed non-stationary hydrologic response. The catchment area ranges from 18.7 to 5158 km² in catchments with non-stationary hydrologic response (Table 1). Therefore, catchment area is not a major factor causing non-stationary response. We also explored the relationships between mean catchment slope (using a 90 m Digital Elevation Model of Australia) and runoff ratio, water balance ET (P-Q), and annual fractional vegetation cover. While increases in runoff ratio, P-Q and mean fractional vegetation cover with increases in mean catchment slope are observed in catchments with nonstationary hydrologic response, no distinct differences between catchments with stationary and non-stationary response are observed.

[Figure]

Top panel shows the relationships between mean catchment slope and runoff ratio, P-Q and mean fractional vegetation cover in catchments with non-stationary hydrologic response. The bottom panel shows the same relationships for catchments that do not exhibit non-stationary behavior.

In section 3.2 of the revised manuscript, the following paragraph is added:

The catchment area ranges from 18.7 to 5158 km$^2$ in catchments with non-stationary hydrologic response (Table 1). While increases in runoff ratio, P-Q and mean fractional vegetation cover with increases in mean catchment slope are observed in catchments with non-stationary hydrologic response, no distinct differences between catchments with stationary and non-stationary hydrologic response are observed.

7. Lack specific data and method descriptions. For example, authors didn't explain how ET and PET were calculated etc.

Thank you for your comment. We included detailed descriptions of ET and PET computations in the revised manuscript.

Section 2.1.: AWAP potential evapotranspiration is calculated based on the Priestley-Taylor equation (Raupach et al., 2009).

Section 4.3: AWAP ET is based on daily transpiration and soil evaporation values obtained from the WaterDyn model that simulates terrestrial water balance across Australia at 5 km resolution.

[revised manuscript text omitted]